# Climate-assisted persistence of tropical fish vagrants in temperate marine ecosystems

Laura Gajdzik [1,2✉], Thomas M. DeCarlo [3], Adam Koziol[1,4], Mahsa Mousavi-Derazmahalleh[1], Megan Coghlan[1], Matthew W. Power[1], Michael Bunce[1,5], David V. Fairclough[6], Michael J. Travers[6], Glenn I. Moore[7,8] & Joseph D. DiBattista [1,9]

Rising temperatures and extreme climate events are propelling tropical species into temperate marine ecosystems, but not all species can persist. Here, we used the heatwave-driven expatriation of tropical Black Rabbitfish (*Siganus fuscescens*) to the temperate environments of Western Australia to assess the ecological and evolutionary mechanisms that may entail their persistence. Population genomic assays for this rabbitfish indicated little genetic differentiation between tropical residents and vagrants to temperate environments due to high migration rates, which were likely enhanced by the marine heatwave. DNA metabarcoding revealed a diverse diet for this species based on phytoplankton and algae, as well as an ability to feed on regional resources, including kelp. Irrespective of future climate scenarios, these macroalgae-consuming vagrants may self-recruit in temperate environments and further expand their geographic range by the year 2100. This expansion may compromise the health of the kelp forests that form Australia's Great Southern Reef. Overall, our study demonstrates that projected favourable climate conditions, continued large-scale genetic connectivity between populations, and diet versatility are key for tropical range-shifting fish to establish in temperate ecosystems.

[1] Trace and Environmental DNA Laboratory, School of Molecular and Life Sciences, Curtin University, Bentley, WA 6102, Australia. [2] Reef Ecology Laboratory, Red Sea Research Center, King Abdullah University of Science and Technology, 23955 Thuwal, Saudi Arabia. [3] College of Natural and Computational Sciences, Hawai'i Pacific University, Honolulu, HI 96744, USA. [4] The GLOBE Institute, Faculty of Health and Medical Sciences, University of Copenhagen, 1017 Copenhagen, Denmark. [5] Institute of Environmental Science and Research, Kenepuru, Porirua 5022, New Zealand. [6] Western Australian Fisheries and Marine Research Laboratories, Department of Primary Industries and Regional Development, Government of Western Australia, North Beach, WA 6920, Australia. [7] Collections and Research, Western Australian Museum, Welshpool, WA 6106, Australia. [8] School of Biological Sciences, University of Western Australia, Nedlands, WA 6907, Australia. [9] Australian Museum Research Institute, Australian Museum, Sydney, NSW 2010, Australia.
✉email: laura.gajdzik@gmail.com

Climate change is affecting the function and composition of nearly all ecosystems on Earth[1,2]. Despite local extirpation, some organisms can respond to gradual rising temperatures by adapting to the warmer conditions, re-distributing to areas with more suitable conditions, or a combination of both[3,4]. In general, terrestrial ectothermic species move to greater altitude, whereas marine species migrate to higher latitude (poleward shift) or into deeper, presumably cooler waters[5,6]. However, the pace of organismal responses to the velocities of ocean warming can be disrupted by extreme climate events. Marine heatwaves, described as anomalous warm-water events that vary in duration, intensity, frequency, and size[7], can reduce biodiversity by subjecting entire ecosystems to thermal stress, leading to species' range contraction or local extinction. Some key examples are die-offs of kelp forests[8], widespread coral bleaching[9], and mass mortality of seabirds, fish, and marine mammals[10]. Conversely, heatwaves associated with oceanic boundary currents have also been reported to increase the geographic distribution of various tropical and subtropical organisms[11,12]. The resulting "tropicalisation" of temperate marine environments is an alarming phenomenon opening new ecological niches for range-shifting organisms, adding extra biotic pressure on native communities[2,13–15].

Despite numerous studies on poleward expatriation of tropical species, few of these explored the ecological and evolutionary mechanisms that may confer resilience in temperate environments. The ephemeral nature of heatwaves may enable species to enter new ecosystems, but whether those species remain after the heatwave depends largely on the thermal sensitivity of fundamental biological traits, resource availability, competition with native species, and a rapid evolution to climate variation[3,16]. Recent empirical evidence indicates a possible matching pace between environmental and evolutionary changes, making it relevant to compare current and projected responses to climate change[17]. Considering the greater occurrence of extreme climate events in the Anthropocene Epoch[7] and the fast niche shifts of introduced species[18], it is thus critical to determine whether marine tropical vagrants will permanently establish in new environments with future warming scenarios.

The west coast of Australia, with its latitudinal gradient in water temperature[2] (Fig. 1a), hosts diverse marine biota from tropical to temperate climates. This coastline experienced the highest category[7] of a marine heatwave in 2010/2011 when a strong La Niña event strengthened the poleward-flowing Leeuwin Current and drove an unusual surge of warm water to more southern, colder locations[19,20] (Fig. 1b). This event accelerated the introduction of numerous early life stage herbivorous animals from tropical waters into temperate environments, notably the Black Rabbitfish *Siganus fuscescens* (Houttuyn 1782)[12,21]. As a result, reproductive adults of *S. fuscescens* were observed in temperate environments at the margin of their previously described geographic distribution[12,22], providing a unique opportunity to investigate the mechanisms that enhance the persistence of vagrant fish populations among temperate communities.

Using a genotyping-by-sequencing approach, we first assessed the population connectivity of *S. fuscescens* collected over four years (2013–2017) across 16° of latitude along the coast of Western Australia (Fig. 1a). Genetic differentiation and the

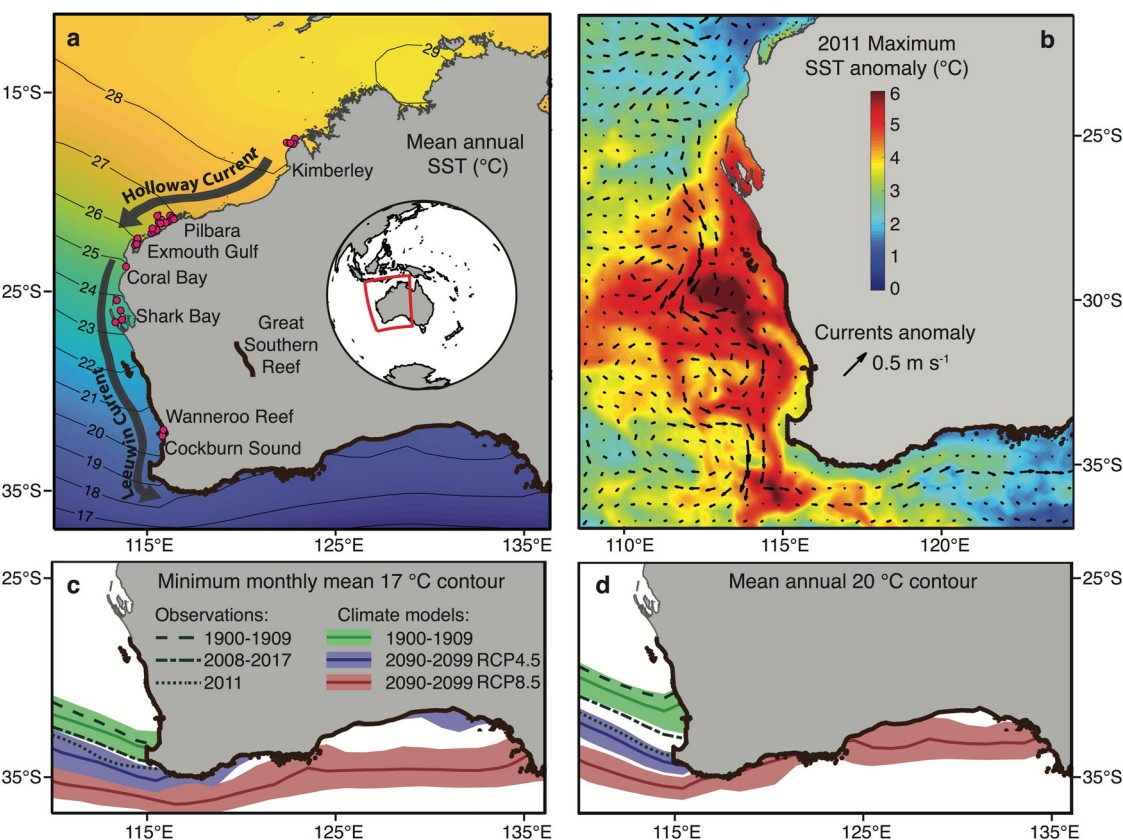

**Fig. 1 Latitudinal gradient in sea surface temperature in Western Australia, the extreme marine heatwave in 2010/2011, and past observations and centurial projections of isotherms according to two CO₂ emission scenarios. a** Map of the west coast of Australia with sampling sites (pink dots) of the Black Rabbitfish *Siganus fuscescens* and mean sea surface temperature (SST) from HadISST using the present-day climatology. **b** The 2010/2011 marine heatwave represented by both the SST (°C) anomaly obtained from NOAA Coral Reef Watch (https://coralreefwatch.noaa.gov/) and near-surface current anomaly (m s⁻¹; arrows). Past observations and projections of the **c** minimum monthly mean 17 °C isotherm and **d** mean annual 20 °C isotherm represented by decadal averages (thick lines) and decadal ranges (shading) from a suite of historical datasets and 11 climate models CMIP5.

degree of gene flow between tropical residents and vagrants were compared to determine whether the 2010/2011 marine heatwave had enhanced the migration of rabbitfish individuals in a southward direction, resulting in panmixia among coastal populations of *S. fuscescens*. We then compared the feeding ecology of tropical residents and vagrants by characterizing their stomach contents with DNA metabarcoding to determine whether rabbitfish can adapt their diet by feeding on novel resources. Finally, we projected isotherms (i.e., contours representing states of equal temperature) based on key life-history trait assumptions to infer whether these range-shifting rabbitfish might overwinter, spawn, and further extend their geographic range into the waters of southern Australia by the end of this century. If future thermal conditions are favourable, vagrancy may lead to permanency with adverse effects for the native temperate marine communities.

## Results and discussion

Our genomic connectivity analysis based on 5,507 putatively neutral single-nucleotide polymorphism (SNP) loci revealed no significant genetic differentiation ($F_{st}$ and $G_{st}$ values) (Supplementary Tables S1 and S2) among most tropical and temperate sampling sites, although two tropical areas, Kimberley and Shark Bay, were slightly more differentiated from the rest (Fig. 2a). Likewise, a Bayesian clustering analysis suggested a $K = 1$ population when considering the neutral SNP dataset (Fig. 2b). However, a second cluster ($K = 2$) was identified for the outlier SNP dataset (Fig. 2b), which could be attributed to sampling bias due to a low number of individuals sampled in Exmouth Gulf and Coral Bay. Alternatively, the presence of a second cluster may indicate some form of adaptive divergence between these tropical sites, which could not be linked to specific genes. Indeed, there

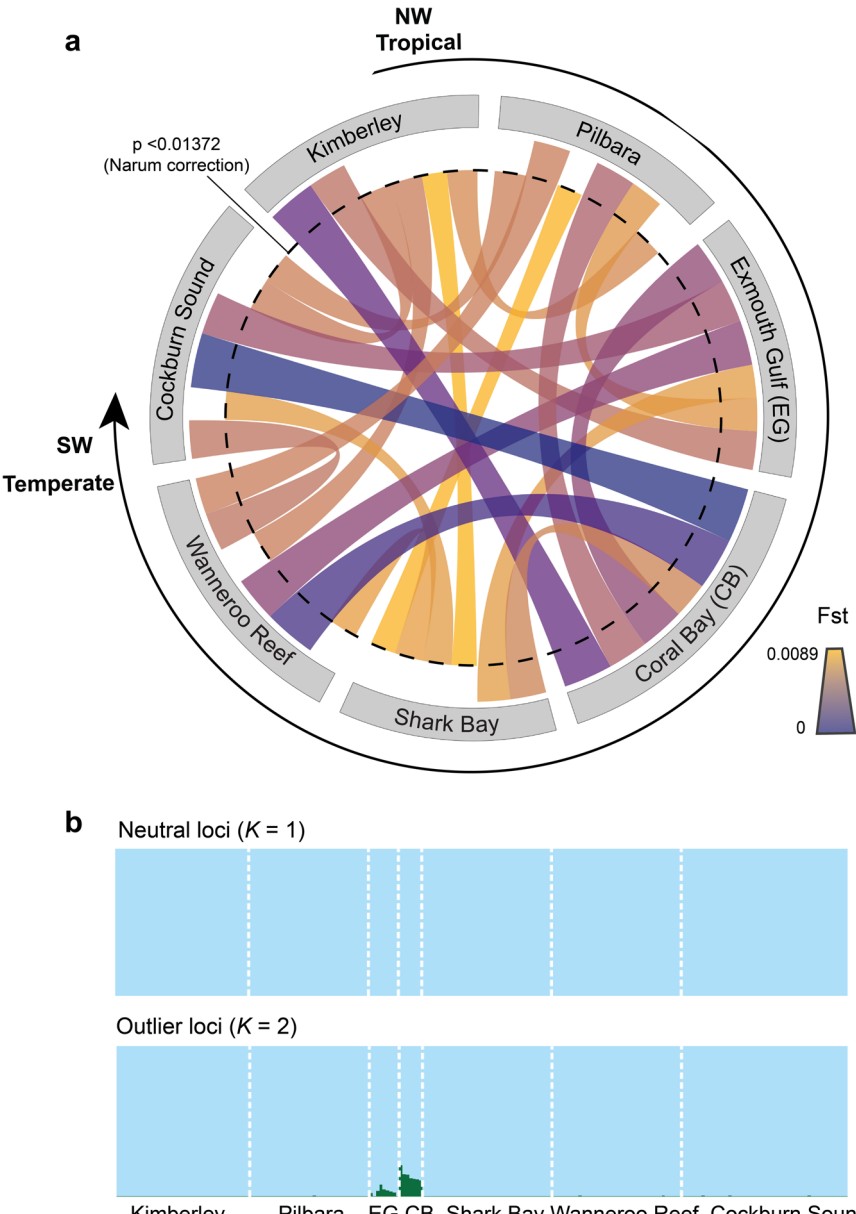

**Fig. 2 Limited evidence of population genetic differentiation among Black Rabbitfish (*Siganus fuscescens*) individuals sampled from northwestern tropical to southwestern temperate marine environments in Australia. a** Circular representation of pairwise genetic differentiation ($F_{st}$ values; fixation index) among rabbitfish individuals sampled from seven sites (spanning tropical to temperate environments) along the west coast of Australia with the significance level indicated by a dashed line. **b** Population stratification among rabbitfish individuals sampled from seven sites along the west coast of Australia ($K = 1$) for neutral SNP loci and ($K = 2$) for outlier SNP loci using fastSTRUCTURE. The colour code denotes population affiliation.

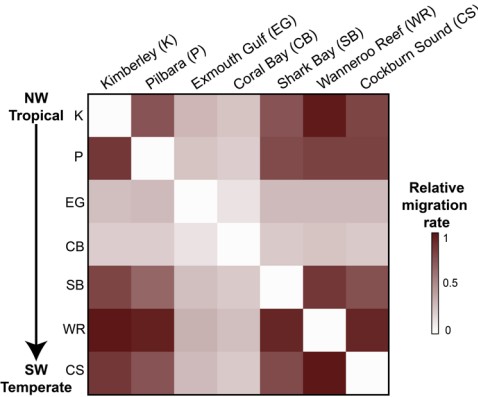

**Fig. 3 Heatmap comparing migration rates among Black Rabbitfish (*Siganus fuscescens*) individuals sampled from northwestern tropical to southwestern temperate environments in Australia.** Columns represent source populations, and rows are receiving populations. No significant asymmetric migration was found.

was no available sequence alignment for the top SNP candidate with heterozygosity >0.1 and *q*-value <0.01. The overall limited genetic differentiation was strengthened by the relatively high degree of gene flow and non-directional migration rates (i.e., equal rates in all directions) between tropical residents and vagrants (Fig. 3 and Supplementary Table S3). The only exceptions were the genetically differentiated tropical sites (Exmouth Gulf and Coral Bay) that appeared to have limited exchange of individuals, indicative of some degree of genetic isolation (Fig. 3). However, the spatial autocorrelation analysis between genetic and geographic distances further confirmed the absence of local-scale genetic structure (Supplementary Fig. S1). Together, these results suggest a large-scale migration of rabbitfish larvae along the coast of Western Australia, which is likely driven by the continuous poleward Leeuwin Current and was probably enhanced during the 2010/2011 marine heatwave. At the time of the heatwave, the Leeuwin Current increased in intensity and flowed unusually early (in summer rather than autumn), dispersing the early life-history stages of summer- and autumn-spawning individual fish further south[12]. The maintenance of above-average water temperatures in this region for the next two years allowed these fish to survive and for some species, like *S. fuscescens*, to exhibit reproductive activity up to five years after the heatwave[12,22]. Although Australia's temperate fish species have historically navigated through range contractions and expansions, they have occurred in response to repeated glacial cycles[23] and thus over evolutionary timescales. In the Anthropocene Epoch, rates of range-shifts, such as that observed for *S. fuscescens*, may be accelerated by the greater climate instability and recurrent extreme events. Yet, the absence of population subdivision in *S. fuscescens* along the coast of Western Australia and the similar levels of genetic diversity (heterozygosity) between vagrants and tropical residents (Supplementary Table S4) indicate a persistent gene flow that could provide sufficient standing genetic variation to promote adaptation in range expansion zones.

For such an evolutionary response to occur, recruiting to new environments as larvae or juveniles needs to include persistence, which depends on the efficiency with which organisms can exploit food resources in novel environments[24]. Our dietary DNA metabarcoding analysis focusing on phytoplankton and algae indicated that rabbitfish individuals from Western Australia have a diverse diet. We identified 78 food items in the stomachs of *S. fuscescens* (Fig. 4 and Supplementary Data 1) that greatly varied in abundance with the majority of the top 30% of sequences

assigned to red and brown macroalgae (Supplementary Fig. S2). These macroalgae differ in geographical distribution, climate affinity, and ecological roles[8,25–27]: they are (i) widespread with records from tropical to temperate environments (e.g., *Asparagopsis taxiformis*, *Champia parvula*, *Lobophora variegata*, *Padina australis*, *Spyridia filamentosa*, and *Tolypiocladia glomerulata*), (ii) mainly tropical (e.g., *Coelothrix irregularis*), (iii) invasive (i.e., *Sargassum natans*), and (iv) habitat-forming (e.g., the kelp *Ecklonia radiata*) (Fig. 4 and Supplementary Data 1). Additionally, epilithic resources, including diatoms (e.g., *Licmophora* sp.), dinoflagellates (i.e., *Dynophysis* sp.), microalgae (e.g., *Gymnochlora* sp.), and cyanobacteria (e.g., *Synechococcus* sp.) were also found in the stomachs of rabbitfish individuals (Fig. 4 and Supplementary Data 2). A multidimensional (nMDS) ordination revealed a limited overlap of the stomach contents of rabbitfish between tropical (Coral Bay and Shark Bay) and temperate regions (Wanneroo Reef and Cockburn Sound); the latter being completely segregated (Fig. 5a). This limited overlap was supported by the permutational multivariate analysis of variance (PERMANOVA; df = 2, F = 6.56, $R^2$ = 0.47, p = 1 × 10^{-5}) and confirmed by the centroid-based homogeneity of group dispersions (PERMDISP2; df = 2, F = 2.43, p = 0.09). The associations between stomach contents and locations were additionally tested using the indicator species analysis (IndVal[28]) and revealed that 18% phytoplankton and algal resources were unique to a region or a combination of regions (Supplementary Data 2). Red macroalgae were the food sources that most uniquely characterized each climate region (e.g., *Gelidiella* sp., *Melanothamnus* sp., *Hydrolithon* sp.) and were also significantly associated between tropical and temperate regions (i.e., *Gayliella* sp. and *Palisada* sp.). Only one microalga (i.e., *Picochlorum* sp.) was uniquely associated with one tropical region, whereas brown algae (order Ectocarpales) and the kelp *E. radiata* significantly characterized the stomach contents of one temperate region (Figs. 4 and 5b). Together, our results emphasized that tropical residents and vagrants exhibited distinct dietary patterns most likely dictated by the latitudinal gradient in available marine biota[13], as observed previously for other range-shifting fish species (e.g., baldchin groper *Choerodon rubescens*)[29]. However, our small sample size for the tropical sites (4–7 individuals) may impede any definite conclusions on local adaptation to the available food resources.

The diverse diet that we detected may also be attributed to a trade-off between nutritional demands, digestive ability, and a tolerance to ingested toxins. Rabbitfish can target cytotoxic terpenoids present in some macroalgae[30], reducing competitive interaction for resources with native temperate fishes (e.g., the Mediterranean Sea[31]). However, the nutritional value of such food sources remains questionable due to reduced digestibility[32]. Additionally, the diet versatility of *S. fuscescens* from Western Australia is likely enabled by a flexible gut microbiome. Jones and colleagues[33] discovered that some of the same individuals used in this study (Supplementary Data 3) possessed a core gut microbiome allowing the continuous digestion of sources along the environmental gradient as well as a specialized hindgut community that regulates the algal fermentation process. Such an optimized diet-microbiome relationship may further facilitate the fast growth of *S. fuscescens*[34] and its success as a vagrant by representing an additional mechanism to withstand seasonal and stochastic fluctuations in environmental resources.

Persisting in new environments also depends on the ability of introduced populations to cope with new climate conditions by either having a wider fundamental niche than the realized niche or by expanding their niche along the temperature axis[18,35]. Our projections of the winter minimum isotherm of 17 °C and the mean annual isotherm of 20 °C—thermal thresholds of *S.*

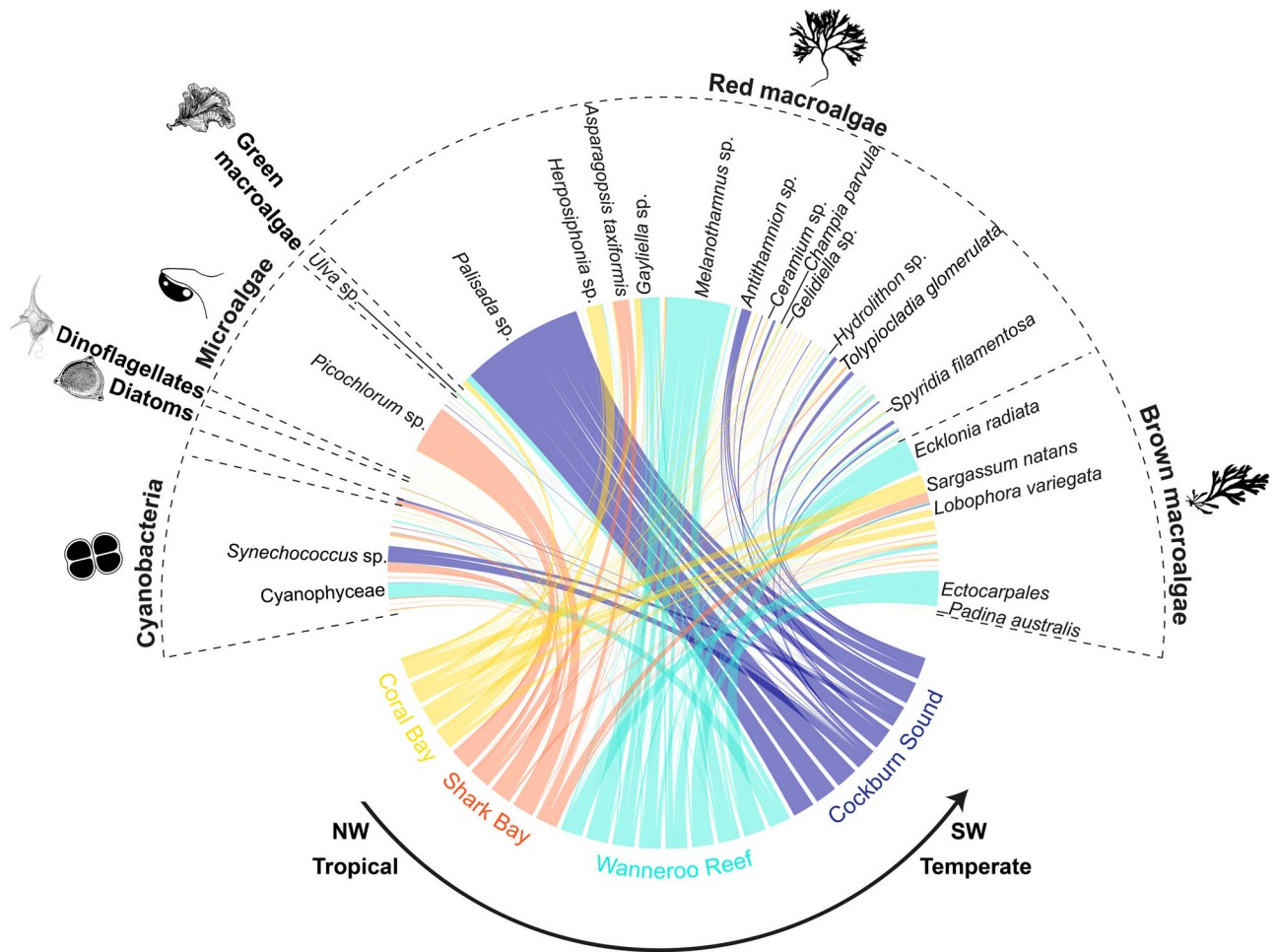

**Fig. 4 Phytoplankton and algal diversity representing the diet of the range-shifting Black Rabbitfish (*Siganus fuscescens*) from northwestern tropical to southwestern temperate marine environments in Australia.** The links between food sources (grouped by phylum) and regions in Western Australia represent the relative abundance of food items per individual. Some key phytoplankton and algal species were mentioned on this plot and the rest of the species names can be found in the Supplementary Data S1.

*fuscescens* for overwintering and spawning[22,36–38]—showed a substantial shift poleward by 2100, irrespective of the carbon emission scenario (Fig. 1c, d). Critically, these contours might encompass the entire southern coast of Australia by 2100 under a very high carbon emission scenario (RCP8.5), but this species would be restricted to the western coast of Australia under an emission stabilization scenario (RCP4.5; Fig. 1c, d). Thus, unless carbon emissions are lowered (RCP4.5), future thermal conditions in southern Australia may allow rabbitfish vagrants to survive and self-recruit ~2,400 km from their post-heatwave geographic range limit, especially considering that the cohort of rabbitfish vagrants mainly consisted of maturing and reproductively mature individuals (Supplementary Data 3). Nevertheless, such a scenario will also depend on the connectivity among patches of seagrass and macroalgal habitats, which are fundamental for the settlement of juvenile rabbitfish[22,34]. For instance, the poleward-flowing East Australian Current expatriates many tropical fish species on a yearly basis into the temperate environments of southeastern Australia, but many of them do not survive due to the cool of winter temperatures and the absence of habitat for recruitment[39]. Likewise, the intensified Leeuwin Current in the heatwave years 2010/2011 in Western Australia resulted in the introduction of new tropical fish species to Rottnest Island, which did not survive during the cooler years that followed[40]. At present, the southern coastline of Australia

consists of a high kelp cover, extensive seagrass meadows, large sand banks, and rocky reefs[41]. The establishment of *S. fuscescens* vagrants will be contingent on the warm-water tolerance of these native macroalgae (e.g., higher productivity of kelps up to 24 °C[8]), the ongoing enzymatic activity in the gut necessary to breakdown non-polysaccharides from macroalgae (e.g., cellulase starts to be inactive at 15 °C[42]), and the vagrants ability to exploit non-macroalgal habitats as nurseries (e.g., turfing algae, sea urchins barrens, or rocky reefs[43]).

Overall, this study supports a possible scenario for the establishment of *S. fuscescens* originally from tropical reefs in Western Australia far into the Great Southern Reef (Fig. 1) based on suitable climate conditions for survival and recruitment, a maintenance of genetic diversity through connectivity, and a versatile diet facilitated by a flexible hindgut microbiome[33]. Much like the Great Barrier Reef, the Great Southern Reef is one of Australia's biodiversity hotspots whose foundation habitat consists of extensive kelp forests[44], which contributed 10 billion dollars (AUD) per year to the gross domestic product of Australia by supporting productive fishing grounds and tourism[8]. The resilience and permanency of macroalgae-consuming vagrants, such as *S. fuscescens*, may form a cumulative impact on the climate-threatened kelp forests, and add to an irreversible restructuring of temperate marine communities on a global scale[8]. Our findings further highlight the fundamental role of

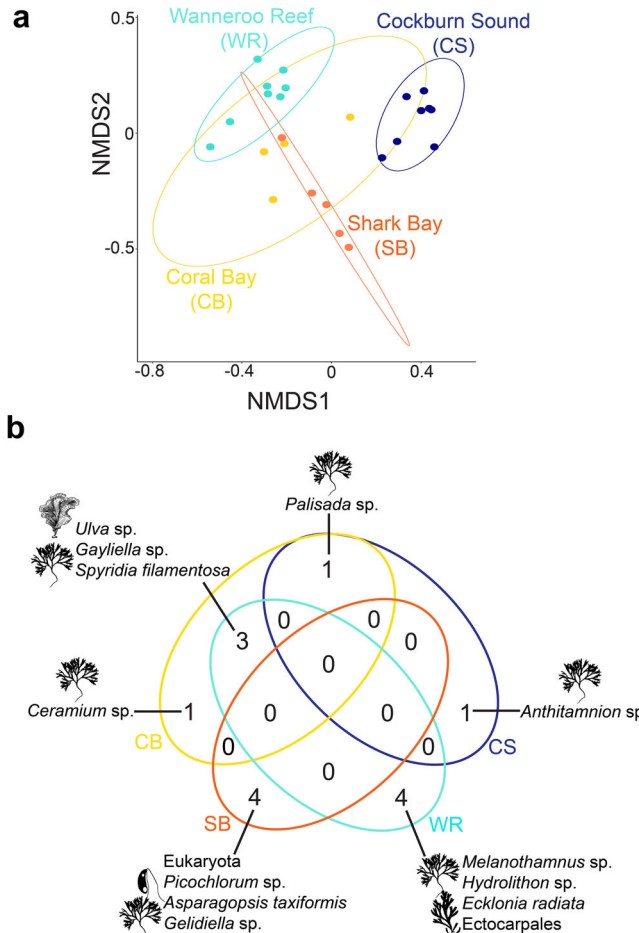

**Fig. 5 Differences in diet composition between Black Rabbitfish (*Siganus fuscescens*) tropical residents (Coral Bay and Shark Bay) and vagrants to temperate marine environments (Wanneroo Reef and Cockburn Sound) from Western Australia. a** Two-dimensional non-metric multidimensional scaling (nMDS) ordination of food items used by rabbitfish individuals from four sites (stress value of 0.12) and grouped by 95% ellipses. **b** Venn diagrams showing the significant number of food sources uniquely characterizing or strongly associated with the stomach contents of *Siganus fuscescens* from a single region or a combination of regions.

large-scale population connectivity and broad dietary and climate characteristics[45] in determining the tropical "winners" in temperate marine range expansion zones.

## Methods

**Population genomics**. DNA was sourced from fin clips or gill tissue sampled from 223 individuals of *Siganus fuscescens* from 2013 to 2017. From the northwest to the southwest of Australia, 40 individuals were sampled from the Kimberley, 36 from the Pilbara, nine from Exmouth Gulf, seven from Coral Bay, 40 from Shark Bay, 51 from Cockburn Sound, and 40 from Wanneroo Reef (Supplementary Data 3). However, following quality filtering of these DNA sequences, three rabbitfish individuals were excluded (see below), resulting in 220 rabbitfish individuals used in all remaining analyses (Supplementary Table S4). These tissue samples were extracted using the DNeasy Blood & Tissue Kit (Qiagen, Germany) based on a modified protocol, which included an in-house binding buffer, 1.4× volume of both wash buffers, and a partial automation of the extractions on a QIAcube (Qiagen) platform to minimize human handling and cross-contamination.

SNP genotyping was conducted using the DArTseq protocol at the Diversity Arrays Technology (University of Canberra, Australia), which is a reduced representation genomic library preparation method that uses two restriction enzymes[46,47]. Genomic DNA was digested with the enzymes *Pst*I–*Sph*I and *Pst*I–*Nsp*I and small fragments (<200 bp) were ligated to adaptors (6–9 bp in length). Polymerase chain reaction (PCR) conditions were as follows: an initial denaturation step at 94 °C for 1 min followed by 30 cycles of 94 °C for 20 s, 58 °C for 30 s and 72 °C for 45 s, with a final extension step at 72 °C for 7 min. After

pooling equimolar PCR products, sequencing was carried out on a single lane of an Illumina Hiseq2500 platform and produced fragments 77 bp in length. Read assembly, quality control, and SNP calling were conducted with the DArT PLD's software DArTsoft14. Additional details about sequence screening, scoring tests, and removal of paralogous sequences are described in DiBattista et al.[48].

Using the R-package *dartR*[49], ~168,000 SNPs were identified in the raw DArT file, which contained 24.71% missing data. From these SNPs, we retained loci genotyped in 95% of individuals and removed loci with coverage <20× and >200× as well as those that were highly variable (heterozygosity >0.75) or rare (allele frequency <0.05). These filtering steps resulted in the retention of 8,366 SNP loci with 1.34% missing data. After removing monomorphic loci and those not present in 90% of individuals (i.e., removal of one individual from Shark Bay, one from Wanneroo Reef, and one from Cockburn Sound), the number of loci was subsequently reduced to 6,505 SNPs across 220 individuals. These loci were then tested for Hardy–Weinberg equilibrium (HWE) and linkage disequilibrium (LD). Loci out of HWE and pairs of loci in LD for at least two populations were removed after Bonferroni correction (i.e., 826 loci), which resulted in 5679 SNP loci. From this dataset, we performed outlier scans between all pairs of sites to identify SNPs that may be under selection using the Outflank method of Lotterhos and Whitlock[50], which is known to result in fewer false positives because it derives the null distribution of population differentiation for neutral loci. Parameters used for Outflank were as follows: (i) 5% left and right trim for the null distribution of $F_{st}$, (ii) minimum heterozygosity for loci of 0.1%, and (iii) a 5% false discovery rate (FDR). After all these filtering steps, our total number of 5,679 SNPs were composed of 5,507 putatively neutral loci and 172 outlier loci, which were separated in "genlight" format for downstream analyses.

**Statistics and reproducibility**. Across these 5,679 SNPs, we calculated the mean allelic richness, the mean expected heterozygosity, and the mean observed heterozygosity using the R-package *diveRsity*[51] and 10,000 permutations[51]. These measures represent the genetic diversity or population's long-term potential for adaptability and persistence[52]. Population genetic differentiation among all sites was then determined by comparing pairwise values of $F_{st}$[53] for the neutral dataset, which were computed with the R-package *StAMPP*[54]. The significance of pairwise $F_{st}$ values was tested using 10,000 permutations via bootstrapping with confidence intervals set to 95% and after correcting for multiple tests using a modified version of the FDR referred to as the BY-FDR[55]. This correction is expected to provide a large increase in power to detect differentiated populations by providing more consistent pairwise significant results relative to the more conservative Bonferroni method[56]. Because the lowest $F_{st}$ was slightly negative (Supplementary Table S1), a constant was added to all $F_{st}$ values such that the minimum $F_{st}$ was 0 and results could be visualized in the circular plot made with the R-package *Circlize*[57]. After transforming the genlight format into a genind format, we also used a second metric of genetic distance ($G_{st}$) to test the robustness of $F_{st}$ values. $G_{st}$ values were computed with the R-package *diveRsity*[51] with 1,000 iterations, which provided similar results with little genetic differentiation among most population pairs (except for Shark Bay) (Supplementary Table S2). We also determined the number of populations in our study using the program fastSTRUCTURE v.1.0 by testing a range of clusters from $K = 2$ to $K = 12$ using default parameters[58] (Supplementary Figs. S3 and S4). The optimum $K$ was obtained using the internal algorithm in fastSTRUCTURE to rank model support and complexity; we determined that population numbers $K = 1$ and $K = 2$ best explained the variation in the neutral and outlier dataset, respectively. Finally, our genetic clusters were further described by visualizing the variation in allelic frequencies between the genotypes using a discriminant analysis of principal components based on 73 retained principle components ($N$ individuals divided by 3) and 12 discriminant functions retained with the R-package *adegenet*[59] ($N − 1$ populations; Supplementary Fig. S5).

Spatial autocorrelation between genetic and geographic distances was also conducted. We assumed that sites further away from each other geographically were more likely to differ genetically, indicative of isolation by distance. Correlation between linearized $F_{st}$ values and "ocean distance" (i.e., geographic distance among sampling sites without crossing land) at depths of 0, 1, and 10 m were conducted using distance-based Mantel tests with the R-package *Vegan*[60] and 100,000 permutations. The ocean distances were extracted from the R-package *marmap*[61]. The scatter plots were produced with the R-package *Vegan*[60] (Supplementary Fig. S1).

To estimate gene flow and direction-relative migration among all sites, we used the divMigrate method[62] implemented in the R-package *diveRsity*[51]. The divMigrate method was selected because it can be directly calculated from standard measures of genetic differentiation and does not need additional parameters to be estimated[62]. This method generates an output with relative migration rates scaled to values between 0 and 1, which is represented in a network that depicts the directionality and migration rates among all sites. These results were based on pairwise $G_{st}$ values[63], which can be found in Supplementary Table S3. However, we also computed the network of migration rates using the Nm[64] and Jost's D values; these results were comparable to those calculated with $G_{st}$ (results not shown). Significant directionalities in gene flow between pairs of populations were also tested with 10,000 bootstrap replications but revealed no significant asymmetric migration. Relative migration rates were visualized using a heatmap, which was computed with the R-package *corrplot*[65].

**Dietary DNA metabarcoding**. From a subset of the individuals used for population genomics, we performed DNA metabarcoding to compare the diet of *S. fuscescens* between 17 tropical/subtropical residents (i.e., seven individuals from Coral Bay and 10 individuals from Shark Bay) and 17 vagrants sampled in temperate environments (i.e., eight individuals from Wanneroo Reef and nine individuals from Cockburn Sound; Supplementary Data 3). The individuals were collected over a period of 4 years (2013–2017) either by hand spear or trap, directly placed on ice and either frozen or processed within 12 h of collection[33]. However, some rabbitfish individuals were excluded due to low sequencing depth or unsuccessful amplification: three individuals from Coral Bay and four from Shark Bay. The resulting low number of fish sampled in the tropical/subtropical locations may therefore not represent the full diet for each population, limiting inferences related to local adaptation to food resources. The goal of this metabarcoding analysis was to specifically identify the marine phytoplankton and algal components of the rabbitfish diet, because this species is known to feed on seaweed (including kelp and seagrass) and cyanobacteria[21,66–68]. The DNA of stomach content samples was extracted using the Mini Stool Kit (Qiagen) following manufacture' protocol, but also including two bead bashing steps (i.e., pre- and post-sample digestion) with ceramic beads using the Tissue Lyzer II (Qiagen).

All extractions were done using between 0.2 and 0.5 g wet weight of stomach content sample, and controls (blanks) were carried out for every set of 12 extractions where all steps remained the same except for the addition of biological material. A portion of the 23S rRNA gene was amplified via PCR using the modified UPA universal primers [p23SrVf1 (5′-GGACAGAAAGACCCTATGAA-3′) and p23SrVr1 (5′-TCAGCCTGTTATCCCTAGAG-3′)], which generates amplicons between 370 and 400 bp in length[69]. Prior to using primers tailed with multiplex identifier (MID) tags that result in unique forward and reverse tag combinations for each sample, PCR tests were performed with different dilutions of DNA extracts (1/5, 1/10, 1/50, and 1/100 dilution in AE buffer) and different annealing temperatures were tested using Applied Biosystems StepOnePlus Real-Time PCR systems. Quantitative PCR (qPCR) reactions were carried out in a total volume of 25 μl with 2 μl of template DNA, 10 μM of forward and reverse primers, 1× AmpliTaq GoldBuffer (Life Technologies, Carlsbad, CA, United States), 2 mM MgCl₂, 0.25 μM dNTPs, 10 μg BSA, 5 pmol of each primer, 0.12× SYBRGreen (Life Technologies), 1 Unit AmpliTaq Gold DNA polymerase (Life Technologies), and Ultrapure Water (Life Technologies). Cycling conditions for these PCRs started with an initial denaturation at 95 °C for 5 min, followed by 35 cycles of 30 s at 95 °C, 30 s at 55 °C, and 45 s at 72 °C, and ended with a final extension for 10 min at 72 °C. The appropriate level of input DNA for metabarcoding (free of inhibition) was determined by using the lowest cycle threshold ($C_T$) value, which is the minimum number of cycles required to exceed fluorescent background levels; this value is inversely proportional to the amount of target DNA. The best amplification curve for each sample had $C_T$ values between 29 and 35. Once the optimal level of input DNA was determined, we performed qPCR using the same volume of reagents and cycling conditions as the optimization reactions, but (i) in duplicate and (ii) with 10 μM of forward and reverse primers that were modified at the 5′ end with a 8 bp-indexing MID tag that resulted in unique forward and reverse tag combinations for each sample. After combining the duplicates of each sample, a maximum of five samples were pooled together based on their $C_T$ values from the qPCRs, and then these new pools were quantified using QIAxcel (Qiagen) to be pooled again to form an equimolar library. This final pooled DNA library was cleaned using a QIAQuick PCR Purification Kit (Qiagen) and quantified as described above.

The ligation of Illumina adapters was achieved with the NEBNext Ultra DNA Library Prep Kit (New England Biolabs, USA) following the manufacturers' protocol and consisted of three main steps: (i) end-repair, (ii) A-tailing, and (iii) ligation of the Illumina phosphorylated adapters. Based on the concentration of DNA in the A-tailed library, we calculated the concentration of Illumina phosphorylated adapters and selected a ratio (insert/vector) of 12/1. The final ligation lasted 1 h, and the ligated library was cleaned with a QIAQuick PCR Purification Kit (Qiagen) but modified by eluting with a miniElute column (Qiagen) in a total volume of 20 μl. The ligated library was then size-selected using a Pippin Prep instrument (Sage Sciences, USA) for fragments between 250 and 600 bp, and subsequently cleaned using a QIAQuick PCR Purification Kit (Qiagen). The final library was quantified using QIAxcel (Qiagen), a high sensitivity kit with a Qubit 4.0 Fluorometer (Invitrogen), and by qPCR with the JetSeq library quantification Lo-ROX kit (Bioline). Sequencing of the 23S rRNA gene was performed on an Illumina Miseq platform (Illumina, USA) housed in the Trace and Environmental DNA (TrEnD) laboratory at Curtin University (Western Australia) using a 500-cycle V2 kit (for paired-end sequencing).

**Statistics and reproducibility**. Paired-end reads were stitched together using the Illumina Miseq analysis software (MiSeq Reporter V.2.5) under the default settings. Sequences were then assigned to samples using MID tag combinations and trimmed in Geneious v.10.2.6 (https://www.geneious.com). To eliminate low-quality sequences, only those with exact matches to sequencing adapters and template-specific primers were kept for downstream analyses. Sequencing reads were dereplicated into unique sequences with USEARCH V.10[70], which was also used to (i) discard sequences with expected error rates >1% and those <100 bp in length, (ii) abundance filter the unique sequences (minimum of two identical reads), and (iii) remove chimeras. Sequences were subsequently clustered into operational

taxonomic units (OTUs) with a cut-off of 97% sequence similarity using USEARCH V.10[70] and normalized to 30,000 reads to allow for cross-sample comparisons. To identify potential sequencing errors, a post-clustering filtering procedure was applied to the original OTU table that contained 1337 OTUS using the R-package *LULU*[71]. The post-clustering algorithm removed potentially erroneous rare OTUs based on both sequence similarity thresholds and within-sample patterns of co-occurrence[71]. To use the LULU algorithm, we had to generate a database of sequences in FASTA format with USEARCH V.10[70] and a match-list of OTUs with similarity score using VSEARCH[72]. Out of the 1,337 OTUs, only 735 OTUs were retained following these clustering steps. Following this, 17 OTUs that appeared in the control samples with at least 2 reads were removed. Prior to their removal, the sequences of these 17 OTUs were blasted against the National Center for Biotechnology Information (NCBI) reference database (using the parameters below) to check whether they matched sequences corresponding to marine phytoplankton and algae, the targets of this study. The taxonomic assignment of the 718 OTUs was performed with the basic local alignment search tool (BLASTn) through the NCBI database using the following parameters: (i) max *E*-value of 0.001, (ii) 100% matching sequence length, (iii) 97% of percentage identity, (iv) a best-hit score edge of 5%, (v) a best-hit overhang of 25%, and (vi) a bit score of >620. OTUs not assigned to bacterial or eukaryotic kingdoms were removed from the dataset and the accuracy of taxonomic assignment was assessed through the use of Australian databases for marine flora and diatoms[25,26]. This resulted in a table containing 86 OTUs, but we only retained OTUs with at least 10 read sequences given that these are less likely to be erroneous sequences that can arise from index-tag jumping. These 78 OTUs—used in downstream statistical analyses—corresponded to cyanobacteria (Cyanophyceae), unknown Eukaryota, dinoflagellates (Dinophyceae), diatoms (Coscinodiscophyceae and Fragilariophyceae), microalgae (microscopic algae of cell size ≤20 μm including Cryptophyceae, Haptophyceae, Mediophyceae, and Chlorarachniophyceae), green macroalgae (Chlorophyta with cell size >20 μm), red macroalgae (Rhodophyta with cell size >20 μm), and brown macroalgae (Ochrophyta with cell size >20 μm) and were represented by silhouettes from PhyloPic (http://phylopic.org/about/) on Figs. 4 and 5, and Supplementary Fig. S2. We then calculated the relative abundance of the 78 OTUs (based on the total number of sequence reads from each individual stomach content, which was visualized in the figure) using a circular plot that was generated with the R-package *Circlize*[57]. We also represented the 30% most abundant OTUs across all stomach content samples with a heatmap using a Bray–Curtis distance matrix, which was computed with the R-package *phyloseq*[73] (Supplementary Fig. S2).

To investigate differences in stomach contents between tropical residents and vagrants to temperate environments, we performed a non-metric multidimensional scaling ordination (nMDS) in two dimensions based on the Bray–Curtis dissimilarity of individuals. The nMDS plot, whose stress value is 0.12, was plotted using the R-package *ggplot2*[74]. To further test the dissimilarity in diet composition among tropical residents and temperate vagrants, a permutational analysis of variance (PERMANOVA) was conducted on the same distance matrix with 100,000 permutations. We also tested the homogeneity of group dispersions using the PERMDISP2 procedure with 100,000 permutations as well. The nMDS plot, PERMANOVA, and PERMDISP2 were done with the R-package *Vegan*[60]. Finally, to highlight food sources that were unique or significantly associated to a single region or a combination of regions, we used the indicator species (IndVal) analysis in the R-package *Indicspecies*[75] with 100,000 permutations and a significance level corrected with the Benjamini and Hochberg (BH) method[76] (Supplementary Data 1 and 2). Significant results were illustrated using colored Venn diagrams on Fig. 5.

The 23S rRNA sequence of the kelp species, *Ecklonia radiata*, from the Western Australian region was not available in the NCBI database, and so three samples were collected in November 2018 at Dunsborough (southwest Australia) and their DNA was extracted with the Miniplant Kit (Qiagen) according to manufacturer's instructions. Prior to extraction, kelp tissues were rinsed with a continuous flow of tap water for 30 min, then soaked in a solution of 70% ethanol, and finally thoroughly rinsed with Milli-Q water. Tissues were also bead-bashed twice with the Tissue Lyzer II (Qiagen) for 30 s on each cycle. The optimal yield of template DNA was estimated with qPCR following the same method as described above. Each kelp sample was prepared for single-step fusion-tag library build using unique index tags following the methods of DiBattista et al.[77] and pooled to form an equimolar library. Size selection was also conducted with a Pippin Prep instrument using the same size range as above, and cleaning was done with QIAQuick PCR purification kit (Qiagen). Final libraries were quantified using a Qubit 4.0 Fluorometer (Invitrogen) and sequenced on the Illumina Miseq platform using 500 cycles and V2 chemistry (for paired-end sequencing).

Paired-end reads were stitched together using the Illumina Miseq analysis software (MiSeq Reporter V. 2.5) under the default settings. Sequences were assigned to samples using MID tag combinations in Geneious v.10.2.6 and reads strictly matching the MID tags, sequencing adapters, and template-specific primers were retained. Each of the three samples was dereplicated into unique sequences. The unique sequence with the highest number of reads (86,000–120,000) was identical in the three samples, and it did not match any 23S rRNA gene sequences available in the NCBI database based on BLASTn. This sequence was thus designated the 23S rRNA voucher sequence of *Ecklonia radiata* from southwestern Australia, blasted against all OTUs found in the stomach of rabbitfish individuals in this study, and deposited on GenBank (accession number MW752516).

**Past and current observations, and climate models**. Historical sea surface temperature (SST) data were acquired from two sources, each with different temporal coverage and spatial resolution. The present-day (2008–2017) and 1900–1909 SST climatologies were calculated from HadISST[78], which is resolved monthly and at 1° spatially. Additionally, the National Oceanic and Atmospheric Administration (NOAA) Coral Reef Watch "CoralTemp v1.0" (daily and 5-km resolution)[79] was used to assess SST anomalies during the 2011 marine heatwave.

Historical and projected SST data were extracted from outputs of a suite of Coupled Model Intercomparison Project Phase 5 (CMIP5) models. We used the monthly-resolution SST model outputs that included historical greenhouse gas (Historical GHG), and representative concentration pathways of 4.5 and 8.5 W m$^{-2}$ forcings ("RCP4.5" and "RCP8.5") runs of the r1i1p1 (designation of initial conditions) ensemble member[80]. These models included ACCESS, CanESM, CMCC, CNRM, CSIRO, GFDL, GISS-E2-H, INMCM, MIROC, MRI, and NorESM[80]. The model SST data for each run (historical GHG, RCP4.5, and RCP8.5) were converted to anomalies relative to a 2008–2017 base period, and these anomalies were added to the HadISST 2008–2017 climatology. This analysis was conducted separately for both mean annual and minimum monthly mean (MiMM). Finally, we calculated ensemble means by averaging the SST anomalies from the 11 models. Ensemble means are plotted in Fig. 1 as decadal averages (thick lines) and decadal ranges (shading) of the mean annual 20 °C contour and the MiMM 17 °C contour. The historical GHG run is used to compare the observed and GHG-forced rates of warming between 1900–1909 and 2018–2019, while the two RCP runs are used to project future (2090–2099) SST scenarios. The observed 1900–1909 contours (from HadISST) fall within the ranges of those from the CMIP5 historical GHG ensembles, indicating that anthropogenic emissions are responsible for warming in this region over the past century.

Surface ocean currents during the 2011 heatwave were assessed using Simple Ocean Data Assimilation (SODA) v.3.3.1[81], a state-of-the-art ocean model constrained by observations when and where they are available. We calculated the near-surface (0–25 m) current anomalies (relative to 1980–2015 mean) for the austral summer (January, February, March, or "JFM") of 2011, which was the peak of the 2010–2011 Western Australia marine heatwave[7]. These current anomalies are plotted on top of SST anomalies in Fig. 1b. All climate analyses were performed in MATLAB2012b.

**Reporting summary**. Further information on research design is available in the Nature Research Reporting Summary linked to this article.

## Data availability

The genetic data are permanently and publicly available in the DRYAD repository at https://doi.org/10.5061/dryad.dr7sqv9xj[82]. MiSeq reads have also been deposited in Genbank: accession numbers PRJNA726752 and MW752516, respectively. Sea surface temperature data are available at https://www.esrl.noaa.gov/psd/data/gridded/data.noaa.oisst.v2.highres.html and https://coralreefwatch.noaa.gov/product/5km/.

## Code availability

R codes for the climate model analysis have been uploaded on Zenodo[83] at https://zenodo.org/record/4535983#.YJg5B2ZKhE5.

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

## Acknowledgements

We are grateful to Heather Dunham, Scott Bennett, Thomas Wernberg, Megan Huggett, Rob Czarnik, Jacquelyn Jones, Sam Moyle, Samuel Payet, Maarten De Brauwer, Jack Parker, and Sue Morrison for providing assistance in the field or in the lab. We also thank the staff of the Aquatic Science and Assessment Division at the Western Australian Department of Primary Industries and Regional Development, particularly Chris Dowling, and the crew of the RV Naturaliste. For molecular guidance in their genotyping service, we thank DArTseq and Andrzej Killian. This work was also supported by resources provided by the Pawsey Super-computing Centre with funding from the Australian Government and the Government of Western Australia. This project was funded by the Wallonia-Brussels Federation via a grant of Excellency for postdoctoral research to L.G. This work was also supported by ARC Linkage Projects (LP160100839 and LP160101508) to J.D.D. and M.B., a Curtin University Early Career Research Fellowship to J.D.D., and a Curtin University award for publication in Science to J.D.D. The Chevron-operated Wheatstone Project's State Environmental Offsets Program funded Pilbara and Gascoyne sample collections as did the Woodside-operated Pluto Project for the State Environmental Offsets Program administered by the Department of Biodiversity, Conservation and Attractions (DBCA). Most fish samples were collected under fisheries exemption permit #2632 issued by the Department of Fisheries WA and the WA Department of Parks and Wildlife (DPAW) license to take fauna for scientific purposes - regulation 17 - (#01-000039-1), and ethically approved (#AEC201527) by the Animal Ethics Committee at Curtin University (Australia). The permit, license, and ethics approvals were issued to J.D.D. Recreational fishers donated the remaining fish samples. The kelp samples were collected under a DPAW - regulation 4 - license (#CE005834) and a WA Fisheries exemption permit (#3321) to Thomas Wernberg (University of Western Australia).

## Author contributions

L.G., J.D.D., and M.B. designed the experiment; J.D.D., G.I.M., D.V.F., and M.J.T. collected fish individuals; L.G., A.L.K., M.C., M.W.P., and J.D.D. performed the lab work; L.G., M.M.-D., and A.L.K. conducted the statistical analysis; T.M.D. created maps and used climate models; L.G. created the population genomics and DNA metabarcoding figures; L.G. wrote the manuscript with substantial contributions of all authors who approved the final version.

## Competing interests

The authors declare no competing interests.
