## [Peer Review File · Communications Biology]

Reviewers' Comments:

Reviewer #1:

Remarks to the Author:

Review of COMMSBIO-20-0841-T; Climate-assisted persistence of tropical fish vagrants in temperate marine ecosystems

This is a straightforward study designed to address two relatively simple questions that have important implications to understand marine population dynamics in the context of climate change: how genetically connected are tropical resident and temperate vagrant fish and do they differ in feeding ecology? The work capitalizes on an ideal biogeographic and oceanographic setting to understand the impacts of climate change in marine population dynamics. The coast of Western Australia (WA) is a hotspot for climate change studies in the sea that includes dramatic examples about how warming events (e.g. marine heatwaves) may result not only in population extinctions, but also on range shifts and colonization of new ecosystems. Previous studies suggest that demes of the study species, the tropical rabbitfish *Siganus fuscescens*, have shifted ranges in association with heatwave(s). The species has apparently also persisted in southern WA reefs during multiple winters. Evidence also suggest that this species has affected habitat structure and increased detritus production in temperate reefs.

The two questions are addressed using a combination of genome-wide (dArT) SNP data from 222 rabbitfish sampled in 13 sites including tropical (i.e. potential source) and temperate (i.e. invaded) sites, as well as dietary DNA metabarcoding data from 39 rabbitfish sampled from two tropical and two temperate sites. The work also includes simplistic results from past observations and centurial projections of isotherms that relate to the thermal threshold for overwintering and spawning for rabbitfish. The latter is an informative addition that helps to contextualize the results for future climatic scenarios.

The study makes novel and substantial contributions to marine ecology. The work is of high quality (but see a potential issue that needs fixing below) and the paper should be of broad interest to a range of scientists, conservation managers and stakeholders.

I listed below a number of issues that probably require clarification and action. Although these include requests for new analyses, it is expected that they will help substantiate further the study conclusions. Such methods represent key analytical components that are missing in what is otherwise a well carried out study.

MAJOR ISSUE:

1 - the spatial analysis of genetic connectivity between tropical and temperate samples is central to the study, but it is currently done using only a few pedestrian tests of population differentiation (e.g. F_{ST} and fastStructure) that strongly suggest a lack of evidence of genetic differentiation. The authors' conclusions on the other hand extend far beyond their analytical findings. These include claims that the lack of evidence supports "continued and enhanced migration..." (lines 96-99) and "persistent gene flow" (i.e. from tropical to temperate regions) (lines 109-111), amongst others. The above conclusions were not tested here and are not necessarily simple to test. That is because determining whether weak or no genetic differentiation is due to a demographically connected metapopulation or is a consequence of the large size of demographically independent populations can be quite challenging in abundant marine species (e.g. see Gagnaire et al., 2015).

I suggest two types of analyses that can potentially help supporting such key claims by objectively testing for different demographic scenarios that could account for observed patterns.

First, comparisons of levels of genetic diversity (it was surprising not to see these summary statistics in the paper!) could be informative given that reductions in genetic diversity are expected to be found in newly colonized regions. Here you could simulate genetic diversity under range expansion to determine whether you have the power to reject certain demographic scenarios (e.g. presence or absence of a founder effect, varying immigration rates to the range extension zone and a range of population growth rates). In fact, given your biogeographic scenario, even the rejection of a founder effect is probably a more informative finding than a $K=1$

from a Structure plot. An example of how this analysis can be achieved in a scenario of high gene flow due to a climate change-driven range expansion (just like rabbitfish) is found in Banks et al. (2010). This was carried out for the highly invasive black sea urchin along Australia's east coast.

Second, a simplistic 'seascape genetics' analysis would add a much needed spatial component to explicitly test for signatures of range expansion in the data. In our sea-urchin paper above we implemented this analysis using population-specific F_{ST} (estimated in GESTE via GLMs). Here, population specific F_{ST} s are expected to be reduced in demes that have rapidly expanded their ranges. That seems to be a very similar scenario as in the rabbitfish.

Other seascape genetic methods that are potentially useful here include spatial autocorrelation analysis using PCA. Below is the relevant Adegenet tutorial; an example of how this test performs in wild populations genotyped with SNPs is in Creel et al. (2019). Also see the recent Maigre et al. (2020).

<http://adegenet.r-forge.r-project.org/files/tutorial-spca.pdf>

Other issues:

2 - Do the temperate samples include different age cohorts (e.g. juveniles, mature individuals?). That would be important information to include in Table S3 and in the main text given that the species apparently failed to recruit in the invaded region by 2013/2014 (Lenanton et al. 2017) and that your temperate samples were obtained after 2015. I assume that size data (or even better, age data derived from otoliths) are available for the individuals listed in Table S3? A relevant explanation about what these data (or in the unlikely event, the lack of it) impacts on the study should also be included.

3 - Dietary analysis section: although this is outside my expertise, it appears that the sample size used (both in number of localities and individual per localities) was rather small compared to other diet studies. Authors should mention the potential limitations and the relevant implications of this issue in the Discussion.

I am also listing below a number of minor issues and relevant references. Overall, I commend the authors on this relatively well executed and interesting study.

Best regards,
Luciano Beheregaray

Minor issues:

- Line 31: change "...to determine the mechanisms bestowing persistence in novel environments" to "... to assess potential mechanisms..." (or something similar). This aim in the Abstract is an overstatement given the study's findings.

- Add "putatively" in line 89 (and elsewhere where "neutral" is used in the context of SNPs) to read:

"based on 5,507 putatively neutral single nucleotide polymorphism (SNP)..." This is needed since this study did not establish the 'neutrally evolving' status of the SNPs, but it instead generated evidence that suggest these might not be under selection.

- Lines 94-95: Replace "some local adaptation" with "adaptive divergence" to read:

"for the outlier dataset emphasizing some local adaptation for two of the tropical sites". This is needed since this study did not establish either functional or experimental evidence for local adaptation based on the outlier candidate SNPs.

- Lines 157-158: clarify if "current geographic range limit" here refers to range before or after the presumed heatwave-induced range shifts:

"survive and self-recruit up to 1,400 km from their current geographic range limit in Western Australia"

- Line 186: include detailed information about individual sizes/age/etc (see comment above).
- Line 197: This is a contentious comment that needs correction; e.g. we use both DArT and ddRAD in my lab, and our DArT data usually have lower coverage than the ddRAD data.
- Line 206-215: provide information in a Table in SI about number of SNPs retained / removed in each filtering step. See Table 2 in Sandoval-Castillo et al. (2018) as example.
- Line 229 and Supplementary Table S5: provide results about the statistical significance of Nei's G_{st} pairwise comparisons.

REFERENCES:

Banks SC, Ling SD, Johnson CR, Piggott MP, Williamson JE & Beheregaray LB (2010) Genetic structure of a recent climate-change driven range expansion. *Molecular Ecology* 19, 2011–2024.

Creel, S., Spong, G., Becker, M. et al. Carnivores, competition and genetic connectivity in the Anthropocene. *Sci Rep* 9, 16339 (2019)

Gagnaire, P. A., Broquet, T., Aurelle, D., Viard, F., Souissi, A., Bonhomme, F., ... Bierne, N. (2015). Using neutral, selected, and hitchhiker loci to assess connectivity of marine populations in the genomic era. *Evolutionary Applications*, 8, 769–786.

Lenanton, R. C. J., Dowling, C. E., Smith, K. A., Fairclough, D. V & Jackson, G. Potential influence of a marine heatwave on range extensions of tropical fishes in the eastern Indian Ocean—Invaluable contributions from amateur observers. *Reg. Stud. Mar. Sci.* 13, 19–31 (2017).

Mairet, TA, Cox, JJ, Weisrock, DW. A spatial genomic approach identifies time lags and historical barriers to gene flow in a rapidly fragmenting Appalachian landscape. *Mol Ecol.* 2020; 29: 673– 685. <https://doi.org/10.1111/mec.15362>

Sandoval-Castillo J, Robinson N, Hart A, Strain L, Beheregaray LB (2018) Seascape genomics reveals adaptive divergence in a connected and commercially important mollusc, the greenlip abalone (*Haliotis laevigata*), along a longitudinal environmental gradient. *Molecular Ecology* 27, 1603-1620.

Reviewer #2:

Remarks to the Author:

General:

Apologies for the delay, these are wild times. Overall, this manuscript is well-written, creative, and meritorious. Blending multiple data types and sampling strategies (including several types of gut content analysis) with geospatial analyses provides a convincing perspective on the importance of poleward fish range expansions, and their correlation with various scenarios of climate change. Statistical analyses are valid and appropriate. The most notable recommendation is to be consistent with sample size statements throughout---some instances were 222 or 223, and others were 220 (see below). By convention, the term 'fishes' is recommended for referring to multiple species, and fish when referring to a single species. Consider omitting Figures S2 and S3, as the resolution renders them as just colored rectangles without observable results. Figure S5 and Table S4 should be included in the main document if space allows. Minor comments are listed below by line number for convenience.

Specific:

Line 57 - change to...marine temperate species

Line 65 - as a transition, I recommend adding a short sentence relating to the mismatch between rapid environmental changes (e.g., bursts of warm water) and slower evolutionary timescales for many endemic/localized species (e.g., hindered by mutation rate, gene flow, selection, drift,

generation time).

Line 74 - animals instead of organisms

Line 76-77 - temperate habitats outside their previously-described geographic distribution

Line 186 - the sample size is listed as 222 above, and 220 in figure 2. Please correct.

Line 217 - fewer false positives because it derives

Line 221 - 222 - it would be interesting to know more about the 172 SNPs that were kept separate. Any idea where they are in the genome, or what their predicted function(s) might be?

Line 226 - Remind me, is a Narum correction similar to the Bonferroni method? Perhaps you could insert parentheses at the end of this sentence to give a quick 'refresher' of what this does...e.g., (reduces the potential impact of incorrect statistical assumptions...?)

Line 244 - Make it more clear in the narrative (abstract and results) whether a subset of the animals were used for gut analyses (a portion of the 222-223), or were these separate individuals. Also, make an obvious statement regarding the total number of gut samples analyzed (e.g., 3 samples per individual and 34 individuals)

Line 256 - delete 'try'

Line 266 - delete second 'of' after input

Reviewer #3:

Remarks to the Author:

In their manuscript, Gajdzik et al assess whether the post El Nina southward expansion of the fish *S. fuscescens* has been facilitated by genetic adaptation or diet flexibility. To this end, they sampled several specimens of the focal species in either tropical/subtropical, or temperate regions of the East coast of Australia. The specimen were subjected to population genomics and diet analysis using DNA metabarcoding. I found the question addressed by this paper overall interesting, with important implications. I have also not much to say about the population structure part of the paper, being not a specialist of the field. However, for the diet part, while I've found the results overall interesting, I also found that the analysis/results presented in this MS were too superficial to fully conclude on the diet versatility of the focal species. I also noticed several methodological issues. These different concerns as well as other minor points are detailed below.

Abstract

#####

The authors are mentioning the microbiota in the abstract, but this is not supported by the data since the microbiota is not directly put in relation with the diet in this study. I would avoid including in the abstract conclusions that are not supported by the data.

Introduction:

#####

The specific questions addressed by this study and their associated working hypotheses could be better explained. At present, only the methodological objectives are mentioned in the introduction, and the relevance of these methods for the particular question addressed and the focal species studied here is unclear.

Fig. 1: panel a) legend for the coral barrier in the middle of the figure along locality names, could be moved to the bottom.

Results/Discussion

#####

l.122ff: it's a bit confusing that widespread species appear to be actually specific to one single site (e.g. Palisada that is mostly found in Cockburn Sound).

Fig.3a: authors should explain the meaning of the thickness of the lines (I guess relative abundance).

l.128ff: These results are over interpreted as it is written: the analysis performed by the authors

does not test for diet differences between the temperate vs. tropical regions, just whether some food items are indicative of a given site. The present results only suggest that part of the diet is site specific, irrespective of the climate. In addition, the current analysis does say whether these specific food items are abundant relative to others in the diet. Diet versatility should be also assessed by accounting for food items relative abundances (e.g. with a PERMANOVA). One other thing also missing is also an assessment of whether these different diets are equivalent from an energetic point of view.

I.142ff: this part on the gut microbiota is too speculative; why not testing explicitly the relationship of the diet with the gut microbiota from the samples analysed here? Data seem to be available, at least partly, from ref. 30. It would be a great addition to the paper.

I.148ff: I have troubles in understanding where the authors are going with this paragraph. The diet versatility point is ok, but one should also discuss about what could be the impact of temperature rise on the distribution of the food too. In other words, will local algae and phytoplankton communities will be the same in the future? Will the native species excluded by competition with *S. fuscescens* or through climate-induced changes in resources?

Methods:

#####

I. 187: for non-specialists, it could be useful here to specify why the authors performed a pluriannual sampling and/or whether this could affect or not the results. Authors could also specify here if the DNA extraction was performed right after each sampling sessions or all at once. Are potential time effects accounted for in the analyses?

The section on population genomics analyses reads ok, but as I said above, I'm not a specialist. However, I could not find the furnish of the restriction enzymes, and the composition of the PCR reactions for that particular section.

The diet analysis section reads difficult because of an extensive use of - sometimes inappropriate - jargon and some unclear descriptions. Some of the methodological choices are also questionable :

I. 249: Stating that one amplifies the bacterial components is misleading because the primer pair used here is largely biased towards algae and cyanobacteria only. Perhaps talking about "phytoplankton and macroalgae" would be less confusing?

I.263: the authors may consider explaining what for MID stands for, this term is not used systematically in the field.

I.263: the marker chosen target a relatively long fragment: 370-400 bp. However, it has been suggested that one should prefer short markers over long ones for diet analyses, since food DNA is highly fractionalized in the gut (reviewed in Taberlet, P., Bonin, A., Coissac, E., & Zinger, L. (2018). Environmental DNA: For biodiversity research and monitoring. Oxford University Press.). In this case, most of the information retrieved with the authors methodology might correspond to material that is actually difficult to digest and hence, of lesser importance in terms of source of energy. Do the authors have any information that would ensure that it is not the case (e.g. a priori known feeding habits based on observations)?

I.267: CT is very specific to the qPCR, it should be explained very briefly.

I.268ff: Are the conditions (mix and thermocycling) the same for the qPCR and for the PCRs sent for sequencing?

I.292: I've read this sentence several times, but it still does not make any sense. Please consider rephrasing by explaining exactly what is the procedure and tools to assemble (i.e. "stitch") the paired-end reads.

I.296: I guess this should read instead that "sequencing reads were dereplicated", or "we kept only one version of replicated reads as well as their associated number of replicates". The term "merge" is inappropriate.

I.297ff: this whole paragraph reads very clumsy, with different things that are lumped together in the same sentence. For example, singletons and chimeras are not removed by performing a clustering at 97% similarity threshold with usearch. The authors removed singletons, chimeras, and also performed a clustering to reduce PCR errors. Hence, the wording is totally misleading. Please consider rephrasing. Likewise, explain what is the rationale for using the lulu algorithm, so that the reader can follow why this step is important and why a matchlist is necessary (and how vsearch is used for that purpose, with what parameters, etc.).

I.304 suggests that the authors removed any sequence occurring in the negative controls. While at first glance, this strategy sounds reasonable, the data produced by the authors may be polluted with "tag-jumps" (see e.g. Esling, P., Lejzerowicz, F., & Pawlowski, J. (2015). Accurate multiplexing and filtering for high-throughput amplicon-sequencing. *Nucleic Acids Research*, 43(5), 2513-2524). This can result in the occurrence of very abundant and biologically meaningful OTUs in negative controls. Did the authors ensure that this bias is limited/absent in their data (e.g. by looking if they obtained sequences with unexpected MID combinations)? Was LULU used to reduce this bias? If none of the above was considered, then removing OTUs in the negative controls could actually remove important OTUs from the dataset.

I.305: What algorithm was used to perform this search (i.e. BLAST or megablast or anything from the blast suite) ? Since the results are a list of matching taxa, how the taxonomic assignment was done exactly? Based on the first/best match? Based on the Last common ancestor of the best matches? How the best matches were defined? Please clarify.

I.309: how this assessment is done? Based on DNA sequences? or Taxon names?

I.310ff the authors explain that they chose to exclude the hindgut and midgut samples from the analysis. Why including these in the MS in the first place then? especially when the methodology is questionable: the DNA extraction protocols are different from that used for the stomach, and even if the authors took cautions to minimise DNA extraction biases effects (standardization of the binding step I.256), this would not remove the most important biases of such an approach, which should be due to the difference in cell lysis efficiency. In addition, they do not support the results obtained from the stomach (Fig S5-7), and these differences are not discussed at all.

I.318: specify how these were standardized.

I.320: The authors included bacterial non-cyanobacterial sequences in their analyses. Unless these are also photosynthetic, I'm not sure these should be considered as part of the diet.

I.321 some Dinophyceae are photosynthetic and therefore part of the phytoplankton. However, they are certainly not diatoms. ^[1]_[SEP]

I.327: The Indval species test is useful to discover indicator species of a given condition, but it is not a proper way to assess whether the diet composition is the same or not in the different locations, it's rather a complementary analysis. The authors should first consider performing a PERMANOVA or a related test (on abundance or presence absence data). In addition, the Indval species metric is also sensitive to rare OTUs (e.g. those occurring in only one sample), are such OTUs included in this analysis?

I.346: "abundance-filtered": what threshold?

Climate-assisted persistence of tropical fish vagrants in temperate marine ecosystems

Laura Gajdzik^{1,2*}, Thomas M. DeCarlo², Adam L. Koziol^{1,3}, Mahsa Mousavi-Derazmahalleh¹, Megan Coghlan¹, Matthew W. Power¹, Michael Bunce¹, David V. Fairclough⁴, Michael J. Travers⁴, Glenn I. Moore^{5,6}, and Joseph D. DiBattista^{1,7}.

We are very grateful to the reviewers for their comments and suggestions that greatly improved the quality of our manuscript (MS). In this revised version of the MS, we have included additional analyses for genetic population connectivity as well as for dietary DNA metabarcoding. We have also addressed concerns regarding potential biases in the DNA metabarcoding methodology. All the minor comments were also taken into consideration and the text was modified accordingly.

Below we respond to the comments of each referee in details. The reviewers' comments are in italics and our response after each point is in normal front.

Reviewer #1

Review of COMMSBIO-20-0841-T; Climate-assisted persistence of tropical fish vagrants in temperate marine ecosystems

This is a straightforward study designed to address two relatively simple questions that have important implications to understand marine population dynamics in the context of climate change: how genetically connected are tropical resident and temperate vagrant fish and do they differ in feeding ecology? The work capitalizes on an ideal biogeographic and oceanographic setting to understand the impacts of climate change in marine population dynamics. The coast of Western Australia (WA) is a hotspot for climate change studies in the sea that includes dramatic examples about how warming events (e.g. marine heatwaves) may result not only in population extinctions, but also on range shifts and colonization of new ecosystems. Previous studies suggest that demes of the study species, the tropical rabbitfish *Siganus fuscescens*, have shifted ranges in association with heatwave(s). The species has apparently also persisted in southern WA reefs during multiple winters. Evidence also suggest that this species has affected habitat structure and increased detritus production in temperate reefs.

The two questions are addressed using a combination of genome-wide (dArT) SNP data from 222 rabbitfish sampled in 13 sites including tropical (i.e. potential source) and temperate (i.e. invaded) sites, as well as dietary DNA metabarcoding data from 39 rabbitfish sampled from two tropical and two temperate sites. The work also includes simplistic results from past observations and centurial projections of isotherms that relate to the thermal threshold for overwintering and spawning for rabbitfish. The latter is an informative addition that helps to contextualize the results for future climatic scenarios.

The study makes novel and substantial contributions to marine ecology. The work is of high quality (but see a potential issue that needs fixing below) and the paper should be of broad interest to a range of scientists, conservation managers and stakeholders.

We thank the reviewer for the positive feedback.

I listed below a number of issues that probably require clarification and action. Although these include requests for new analyses, it is expected that they will help substantiate further the study conclusions. Such methods represent key analytical components that are missing in what is otherwise a well carried out study.

In this revised version of the MS, we have addressed the reviewer's concerns about the lack of analysis for the population genetics part by adding several tests. Please find our detailed reply in the next sections.

MAJOR ISSUE:

1 - the spatial analysis of genetic connectivity between tropical and temperate samples is central to the study, but it is currently done using only a few pedestrian tests of population differentiation (e.g. F_{ST} and fastStructure) that strongly suggest a lack of evidence of genetic differentiation. The authors' conclusions on the other hand extend far beyond their analytical findings. These include claims that the lack of evidence supports "continued and enhanced migration..." (lines 96-99) and "persistent gene flow" (i.e. from tropical to temperate regions) (lines 109-111), amongst others. The above conclusions were not tested here and are not necessarily simple to test. That is because determining whether weak or no genetic differentiation is due to a demographically connected metapopulation or is a consequence of the large size of demographically independent populations can be quite challenging in abundant marine species (e.g. see Gagnaire et al., 2015). I suggest two types of analyses that can potentially help supporting such key claims by objectively testing for different demographic scenarios that could account for observed patterns.

First, comparisons of levels of genetic diversity (it was surprising not to see these summary statistics in the paper!) could be informative given that reductions in genetic diversity are expected to be found in newly colonized regions.

In the revised version of the MS, we now provide a table that includes several genetic measures: mean allelic richness, and mean expected and observed heterozygosity. This information is now available in Table S4 and lines 131-133, and revealed similar levels of H_e and H_o for both vagrants to temperate environments and tropical residents.

Here you could simulate genetic diversity under range expansion to determine whether you have the power to reject certain demographic scenarios (e.g. presence or absence of a founder effect, varying immigration rates to the range extension zone and a range of population growth rates). In fact, given your biogeographic scenario, even the rejection of a founder effect is probably a more informative finding than a $K=1$ from a Structure plot. An example of how this analysis can be achieved in a scenario of high gene flow due to a climate change-driven range expansion (just like rabbitfish) is found in Banks et al. (2010). This was carried out for the highly invasive black sea urchin along Australia's east coast.

We thank the reviewer for suggesting to add a migration test. Instead of using the program SPLATCHE that was applied in Banks *et al.* (2010), we used the function `divMigrate` in the R-package *diveRsity* that represents relative migration levels between population samples (Keenan et al. 2013; Sundqvist et al. 2016). Our results indicate a relatively high degree of gene flow between tropical rabbitfish residents and vagrants to temperate environments, further highlighting a migration of individuals between the tropics and temperate environments (Fig. 3 and Table S3) and please see lines 111-116 and 303-315.

Second, a simplistic ‘seascape genetics’ analysis would add a much needed spatial component to explicitly test for signatures of range expansion in the data. In our sea-urchin paper above we implemented this analysis using population-specific F_{ST} (estimated in GESTE via GLMs). Here, population specific F_{ST} s are expected to be reduced in demes that have rapidly expanded their ranges. That seems to be a very similar scenario as in the rabbitfish. Other seascape genetic methods that are potentially useful here include spatial autocorrelation analysis using PCA. Below is the relevant Adegenet tutorial; an example of how this test performs in wild populations genotyped with SNPs is in Creel et al. (2019). Also see the recent Maigre et al. (2020). <http://adegenet.r-forge.r-project.org/files/tutorial-spca.pdf>
We thank the reviewer for this suggestion. We have performed such analysis using multiple regression tests and found no correlation between geographic and genetic distances. We have now added these results at lines 116-121, described how we performed them in the M&M at lines 296- 302, and added a supplementary figure (Fig. S1).

Other issues:

2 - Do the temperate samples include different age cohorts (e.g. juveniles, mature individuals?). That would be important information to include in Table S3 and in the main text given that the species apparently failed to recruit in the invaded region by 2013/2014 (Lenanton et al. 2017) and that your temperate samples were obtained after 2015. I assume that size data (or even better, age data derived from otoliths) are available for the individuals listed in Table S3? A relevant explanation about what these data (or in the unlikely event, the lack of it) impacts on the study should also be included.

Rabbitfish individuals collected from temperate environments (Wanneroo Reef and Cockburn Sound) were a mixed cohort of maturing and reproductively mature adults of both sexes (male and female), which further sustains the possibility of recruitment of these vagrants in the new temperate environment (see Table S7). This information has been added in lines 193-194. Unfortunately, information about exact age from otoliths is not available for these fish.

3 - Dietary analysis section: although this is outside my expertise, it appears that the sample size used (both in number of localities and individual per localities) was rather small compared to other diet studies. Authors should mention the potential limitations and the relevant implications of this issue in the Discussion.

Our sample size of 7-10 individuals per locality is similar to other diet studies that also used DNA metabarcoding (e.g., Brandl et al. 2020, Takahasi *et al.* 2020).

I am also listing below a number of minor issues and relevant references. Overall, I commend the authors on this relatively well executed and interesting study.

We thank the reviewer for the constructive feedback.

Best regards,
Luciano Beheregaray

Minor issues:

- Line 31: change “...to determine the mechanisms bestowing persistence in novel environments” to “... to assess potential mechanisms...” (or something similar). This aim in the Abstract is an overstatement given the study’s findings.

Done.

- Add “putatively” in line 89 (and elsewhere where “neutral” is used in the context of SNPs) to read: “based on 5,507 putatively neutral single nucleotide polymorphism (SNP)...”. This is needed since this study did not establish the ‘neutrally evolving’ status of the SNPs, but it instead generated evidence that suggest these might not be under selection.

Done.

- Lines 94-95: Replace “some local adaptation” with “adaptive divergence” to read: “for the outlier dataset emphasizing some local adaptation for two of the tropical sites”. This is needed since this study did not establish either functional or experimental evidence for local adaptation based on the outlier candidate SNPs.

Done.

- Lines 157-158: clarify if “current geographic range limit” here refers to range before or after the presumed heatwave-induced range shifts: “survive and self-recruit up to 1,400 km from their current geographic range limit in Western Australia”

Done.

- Line 186: include detailed information about individual sizes/age/etc (see comment above).

Done. Please see table S7.

- Line 197: This is a contentious comment that needs correction; e.g. we use both DArT and ddRAD in my lab, and our DArT data usually have lower coverage than the ddRAD data.

The contentious part of the sentence has been removed.

- Line 206-215: provide information in a Table in SI about number of SNPs retained/removed in each filtering step. See Table 2 in Sandoval-Castillo et al. (2018) as example.

We have already provided the number of SNPs after each filtering step in the main text and prefer not to add another supplementary table. However, we have modified the text in order to bring some clarity to this metric. Please see lines 250-267.

- Line 229 and Supplementary Table S5: provide results about the statistical significance of Nei’s G_{ST} pairwise comparisons.

Done.

REFERENCES:

Banks SC, Ling SD, Johnson CR, Piggott MP, Williamson JE & Beheregaray LB (2010) Genetic structure of a recent climate-change driven range expansion. *Molecular Ecology* 19, 2011–2024.

Creel, S., Spong, G., Becker, M. et al. Carnivores, competition and genetic connectivity in the Anthropocene. *Sci Rep* 9, 16339 (2019)

Gagnaire, P. A., Broquet, T., Aurelle, D., Viard, F., Souissi, A., Bonhomme, F., ... Bierne, N. (2015). Using neutral, selected, and hitchhiker loci to assess connectivity of marine populations in the genomic era. *Evolutionary Applications*, 8, 769–786.

Lenanton, R. C. J., Dowling, C. E., Smith, K. A., Fairclough, D. V & Jackson, G.

Potential influence of a marine heatwave on range extensions of tropical fishes in the eastern Indian Ocean—Invaluable contributions from amateur observers. *Reg. Stud. Mar. Sci.* 13, 19–31 (2017).

Maigret, TA, Cox, JJ, Weisrock, DW. A spatial genomic approach identifies time lags and historical barriers to gene flow in a rapidly fragmenting Appalachian landscape. *Mol Ecol.* 2020; 29: 673– 685. <https://doi.org/10.1111/mec.15362>

Sandoval-Castillo J, Robinson N, Hart A, Strain L, Beheregaray LB (2018) Seascape genomics reveals adaptive divergence in a connected and commercially important mollusc, the greenlip abalone (*Haliotis laevis*), along a longitudinal environmental gradient. *Molecular Ecology* 27, 1603-1620.

Reviewer #2

General:

Apologies for the delay, these are wild times. Overall, this manuscript is well-written, creative, and meritorious. Blending multiple data types and sampling strategies (including several types of gut content analysis) with geospatial analyses provides a convincing perspective on the importance of poleward fish range expansions, and their correlation with various scenarios of climate change. Statistical analyses are valid and appropriate.

We thank the reviewer for the positive feedback.

The most notable recommendation is to be consistent with sample size statements throughout--some instances were 222 or 223, and others were 220 (see below).

The number of individuals has been corrected throughout the main text and detailed information is available at lines 227-232.

By convention, the term 'fishes' is recommended for referring to multiple species, and fish when referring to a single species.

The reviewer is correct, apologies for this oversight. We have modified the text. Please see lines 198, 232, 324, and 452 as examples.

Consider omitting Figures S2 and S3, as the resolution renders them as just colored rectangles without observable results.

We have removed these figures.

Figure S5 and Table S4 should be included in the main document if space allows.

In this revised version of the MS, we have added the information from the original Figure 5 and Table S4 into two new figures presented in the main text: Figure 4 and Figure 5. We still provide the details of these results in the Supplementary Tables S5-S6.

Minor comments are listed below by line number for convenience.

Specific:

Line 57 - change to...marine temperate species

We have now added the missing word to the text.

Line 65 - as a transition, I recommend adding a short sentence relating to the mismatch between rapid environmental changes (e.g., bursts of warm water) and slower evolutionary timescales for many endemic/localized species (e.g., hindered by mutation rate, gene flow, selection, drift, generation time).

We added a transition (see lines 67-69), but emphasized a possible match between a rapid environmental change and fast evolutionary response.

Line 74 - animals instead of organisms

Done.

Line 76-77 - temperate habitats outside their previously-described geographic distribution

Done.

Line 186 - the sample size is listed as 222 above, and 220 in figure 2. Please correct.

We have corrected the number individuals throughout the text.

Line 217 - fewer false positives because it derives

Done.

Line 221 - 222 - it would be interesting to know more about the 172 SNPs that were kept separate. Any idea where they are in the genome, or what their predicted function(s) might be?

Out of the 172 outlier loci, we blasted the entire DNA sequence of the top candidate loci (heterozygosity > 0.1, q-value < 0.01) that were flagged as being under selection. However, the sequence could not be reliably attributed to any specific genes. This result has been indicated in lines 108-111. Alternatively, the presence of a second population for Coral Bay and Exmouth Gulf might be a sample size issue given the low number of individuals for each region (7-8 individuals) compared to other regions in Western Australia (40-50 individuals). Low sampling size is known to induce less precise estimate of allele frequencies, influencing the log-marginal likelihood lower bound (LLBO) and rendering it less reliable in choosing the model complexity (Raj, Stephen, & Pritchard 2014). We have added this information in lines 107-108.

Line 226 - Remind me, is a Narum correction similar to the Bonferroni method? Perhaps you could insert parentheses at the end of this sentence to give a quick 'refresher' of what this does...e.g., (reduces the potential impact of incorrect statistical assumptions...?)

We have added additional info about the correction whose value is from Narum's paper (2006). Please see lines 274-278.

Line 244 - Make it more clear in the narrative (abstract and results) whether a subset of the animals were used for gut analyses (a portion of the 222-223), or were these separate individuals. Also, make an obvious statement regarding the total number of gut samples analyzed (e.g., 3 samples per individual and 34 individuals)

We have now clarified the information in the text (lines 318-326) and provided the exact number of for each next-generation sequencing techniques in Table S7.

Line 256 - delete 'try'

Done.

Line 266 - delete second 'of' after input
Done.

Reviewer #3

In their manuscript, Gajdzik et al assess whether the post El Nina southward expansion of the fish *S. fuscescens* has been facilitated by genetic adaptation or diet flexibility. To this end, they sampled several specimens of the focal species in either tropical/subtropical, or temperate regions of the East coast of Australia. The specimen were subjected to population genomics and diet analysis using DNA metabarcoding. I found the question addressed by this paper overall interesting, with important implications. I have also not much to say about the population structure part of the paper, being not a specialist of the field. However, for the diet part, while I've found the results overall interesting, I also found that the analysis/results presented in this MS were too superficial to fully conclude on the diet versatility of the focal species. I also noticed several methodological issues. These different concerns as well as other minor points are detailed below.

We thank the reviewer for the positive feedback. In the revised version of the MS, we have addressed the reviewer's concerns by adding additional statistical tests and clarified the methods used for DNA metabarcoding.

Abstract

#####

The authors are mentioning the microbiota in the abstract, but this is not supported by the data since the microbiota is not directly put in relation with the diet in this study. I would avoid including in the abstract conclusions that are not supported by the data.

We have removed the sentence about the flexible microbiota from the abstract.

Introduction:

#####

The specific questions addressed by this study and their associated working hypotheses could be better explained. At present, only the methodological objectives are mentioned in the introduction, and the relevance of these methods for the particular question addressed and the focal species studied here is unclear.

The reviewer was correct, apologies for the oversight. We have added hypotheses for each objective. Please see lines 85-98.

Fig. 1: panel a) legend for the coral barrier in the middle of the figure along locality names, could be moved to the bottom.

We believe that the reviewer is referring to the dark brown line representing the Great Southern Reef (GSR). Since the GSR is plotted on the first panel of Figure 1 and there is no extra space to move the legend somewhere else on that same panel, we prefer not to modify the original figure.

Results/Discussion

#####

l.122ff: it's a bit confusing that widespread species appear to be actually specific to one single site (e.g. *Palisada* that is mostly found in Cockburn Sound).

There are 4 *Palisada* species recorded in Western Australia: *P. concreta*, *P. cruciata*, *P. parvipapillata*, and *P. perforata* (<https://florabase.dpaw.wa.gov.au/search/quick?q=palisada>). Some of these species have only been found in the northwestern regions (Kimberley and Pilbara), whereas others are ubiquitous along the coast of western Australia. Given this difference in geographic distribution and the inability to assign the sequence to one of these particular species, we have removed *Palisada* sp. from the line where we highlighted some of cosmopolitan species (line 143).

Fig.3a: authors should explain the meaning of the thickness of the lines (I guess relative abundance).

The reviewer is correct. The arrows represent the relative abundance of each food item that was taxonomically assigned in the stomach of each rabbitfish individual. We have added this information in the figure legend at lines 755-757.

l.128ff: These results are over interpreted as it is written: the analysis performed by the authors does not test for diet differences between the temperate vs. tropical regions, just whether some food items are indicative of a given site. The present results only suggest that part of the diet is site specific, irrespective of the climate. In addition, the current analysis does say whether these specific food items are abundant relative to others in the diet. Diet versatility should be also assessed by accounting for food items relative abundances (e.g. with a PERMANOVA).

We have added additional statistical tests to be able to better interpret differences in diet data. We have performed a nMDS, a PERMANOVA, and a PERMDISP test. The new results have been added in the main text in lines 151-159 and Figure 5 a. The methodologies can be found at lines 417-425.

One other thing also missing is also an assessment of whether these different diets are equivalent from an energetic point of view.

We have added a comment about this at lines 169-174.

l.142ff: this part on the gut microbiota is too speculative; why not testing explicitly the relationship of the diet with the gut microbiota from the samples analysed here? Data seem to be available, at least partly, from ref. 30. It would be a great addition to the paper.

We have added which individuals had the microbiome data in our metadata table (Table S7) as well as modified the sentence at lines 175-178. However, we did not perform these additional tests since Jones *et al.* (2018) already described the microbiota of rabbitfish from western Australia at both the tropical and temperate sites.

l.148ff: I have troubles in understanding where the authors are going with this paragraph. The diet versatility point is ok, but one should also discuss about what could be the impact of temperature rise on the distribution of the food too. In other words, will local algae and phytoplankton communities will be the same in the future? Will the native species excluded by competition with *S. fuscescens* or through climate-induced changes in resources?

The impact of rising temperatures on native food sources and on the feeding behaviour of the rabbitfish are mentioned in lines 204-209. Considering that we can only speculate as to what the phytoplankton community would be composed of in the future, we did not discuss this further, although we do recognize this would be an interesting question to address in future experimental manipulation experiments.

Methods:

#####

l. 187: for non-specialists, it could be useful here to specify why the authors performed a pluriannual sampling and/or whether this could affect or not the results. Authors could also specify here if the DNA extraction was performed right after each sampling sessions or all at once. Are potential time effects accounted for in the analyses?

The sampling years were added in Table S7 (where the metadata for each individual is provided) as well as in lines 322-324. However, seasonality of sampling has not been recorded systematically for all individuals. The goal of this MS is to compare the feeding ecology of pre- and post-heatwave rabbitfish individuals, and our four-year sampling design allowed us to answer (lines 85-87).

The section on population genomics analyses reads ok, but as I said above, I'm not a specialist. However, I could not find the furnisher of the restriction enzymes, and the composition of the PCR reactions for that particular section.

The DArTseq protocol was conducted by the team at Diversity Arrays Technology at the University of Canberra (Australia). Information of the enzyme provider is therefore not known, but protocol details can be found on their website (<http://www.diversityarrays.com/dart-application-dartseq>).

The diet analysis section reads difficult because of an extensive use of - sometimes inappropriate - jargon and some unclear descriptions. Some of the methodological choices are also questionable:

The diet analysis section has been entirely rewritten and now includes less jargon. We conducted a thorough review of metabarcoding approaches and our choices are tested and justified in previously published papers (e.g., West *et al.* 2020; Koziol *et al.* 2018, DiBattista *et al.* 2018). We are confident that our rewritten methods now provide clarity as to our choices (lines 318-453).

l. 249: Stating that one amplifies the bacterial components is misleading because the primer pair used here is largely biased towards algae and cyanobacteria only. Perhaps talking about "phytoplankton and macroalgae" would be less confusing?

We have modified this section but this primer also targets the 23 S rRNA of microalgae, not only of phytoplankton and macroalgae. Please see line 327 for clarification of this point.

l.263: the authors may consider explaining what for MID stands for, this term is not used systematically in the field.

Done. Please see lines 337-338.

l.263: the marker chosen target a relatively long fragment: 370-400 bp. However, it has been suggested that one should prefer short markers over long ones for diet analyses, since food DNA is highly fractionalized in the gut (reviewed in Taberlet, P., Bonin, A., Coissac, E., & Zinger, L. (2018). *Environmental DNA: For biodiversity research and monitoring*. Oxford University Press.). In this case, most of the information retrieved with the authors methodology might correspond to material that is actually difficult to digest and hence, of lesser importance in terms of source of energy. Do the authors have any information that would ensure that it is not the case (e.g. a priori known feeding habits based on observations)?

Using shorter primers, such as trnL, usually only allow broad taxonomic classification (phyla level or family level), whereas longer regions allow for improved taxonomic classification, which is key for our MS that investigates fine-scale dietary patterns. Additionally, S.

fuscescens is known to consume brown (e.g., *Sargassum* spp., *Ecklonia radiata*) and red macroalgae (e.g., *Dictyota* spp.) as well as cyanobacteria (Egan *et al.* 2013; Noda *et al.* 2014; Zarco-Perello *et al.* 2019). Therefore, our choice of primers needed to be able to address this diet diversity.

1.267: CT is very specific to the qPCR, it should be explained very briefly.

Explanation are provided in lines 348-352.

1.268ff: Are the conditions (mix and thermocycling) the same for the qPCR and for the PCRs sent for sequencing?

All PCRs were RT-PCRs and had the same thermocycling conditions (lines 345-347).

1.292: I've read this sentence several times, but it still does not make any sense. Please consider rephrasing by explaining exactly what is the procedure and tools to assemble (i.e. "stitch") the paired-end reads.

This sentence has been rephrased. Please see 376-377.

1.296: I guess this should read instead that "sequencing reads were dereplicated", or "we kept only one version of replicated reads as well as their associated number of replicates". The term "merge" is inappropriate.

This sentence has been rephrased. Please see 378-384.

1.297ff: this whole paragraph reads very clumpy, with different things that are lumped together in the same sentence. For example, singletons and chimeras are not removed by performing a clustering at 97% similarity threshold with usearch. The authors removed singletons, chimeras, and also performed a clustering to reduce PCR errors. Hence, the wording is totally misleading. Please consider rephrasing. Likewise, explain what is the rationale for using the lulu algorithm, so that the reader can follow why this step is important and why a matchlist is necessary (and how vsearch is used for that purpose, with what parameters, etc.).

This paragraph has been modified entirely. The justification for the use of the LULU curation has been added as well. Please see lines 386-392.

1.304 suggests that the authors removed any sequence occurring in the negative controls. While at first glance, this strategy sounds reasonable, the data produced by the authors may be polluted with "tag-jumps" (see e.g. Esling, P., Lejzerowicz, F., & Pawlowski, J. (2015). Accurate multiplexing and filtering for high-throughput amplicon-sequencing. *Nucleic Acids Research*, 43(5), 2513-2524). This can result in the occurrence of very abundant and biologically meaningful OTUs in negative controls. Did the authors ensure that this bias is limited/absent in their data (e.g. by looking if they obtained sequences with unexpected MID combinations)? Was LULU used to reduce this bias? If none of the above was considered, then removing OTUs in the negative controls could actually remove important OTUs from the dataset.

The sequences of the OTUs appearing in the control samples have been blasted and assigned to sequences that belonged to rice and terrestrial plants. We have provided additional details on these assignments at lines 393-397. Additionally, this is common practice in DNA metabarcoding paper to remove sequences from the negative control (e.g., West *et al.* 2020).

1.305: What algorithm was used to perform this search (i.e. BLAST or megablast or anything from the blast suite)? Since the result s a list of matching taxa, how the taxonomic assignment was done exactly? Based on the first/best match? Based on the Last common ancestor of the best matches? How the best matches were defined? Please clarify.

We used BLASTn and the parameters have already been detailed in the original version of the MS (lines 397-401).

1.309: how this assessment is done? Based on DNA sequences? or Taxon names?

It was done using BLASTn as previously described.

1.310ff the authors explain that they chose to exclude the hindgut and midgut samples from the analysis. Why including these in the MS in the first place then? especially when the methodology is questionable: the DNA extraction protocols are different from that used for the stomach, and even if the authors took cautions to minimise DNA extraction biases effects (standardization of the binding step 1.256), this would not remove the most important biases of such an approach, which should be due to the difference in cell lysis efficiency. In addition, they do not support the results obtained from the stomach (Fig S5-7), and these differences are not discussed at all.

We have removed the hindgut and midgut results from this study.

1.318: specify how these were standardized.

This sentence has been rephrased (lines 411-414).

1.320: The authors included bacterial non-cyanobacterial sequences in their analyses. Unless these are also photosynthetic, I'm not sure these should be considered as part of the diet.

The reviewer makes a good point. We have removed the two OTUs that were taxonomically assigned to bacteria and Cytophagales, which are non-photosynthetic bacteria probably part of the rabbitfish's microbiota. Figure 4 representing the diet of rabbitfish has been modified and the statistical analyses rerun to reflect this change.

1.321 some Dinophyceae are photosynthetic and therefore part of the phytoplankton.

However, they are certainly not diatoms.

We apologize for the oversight. We have modified Figure 4 representing the diet of rabbitfish to correct this mistake.

1.327: The Indval species test is useful to discover indicator species of a given condition, but it is not a proper way to assess whether the diet composition is the same or not in the different locations, it's rather a complementary analysis. The authors should first consider performing a PERMANOVA or a related test (on abundance or presence absence data). In addition, the Indval species metric is also sensitive to rare OTUs (e.g. those occurring in only one sample), are such OTUs included in this analysis?

As mentioned before, we have performed three additional statistical analyses to address these issues: nMDS, PERMANOVA, and PERMDISP2.

1.346: "abundance-filtered": what threshold?

This sentence has been rephrased and "abundance-filtered" has now been removed (lines 444-448)

References

- Brandl, S.J., Casey, J.M., & Meyer, C.P. (2020) Dietary and habitat niche partitioning in congeneric cryptobenthic reef fish species. *Coral Reefs*, 39: 305–317.
- DiBattista, J.D., Reimer, J.D., Stat, M., Masucci, G.D., Biondi, P., De Brauwer, M., & Bunce, M. 2019. Digging for DNA at depth: rapid universal metabarcoding surveys (RUMS) as a tool to detect coral reef biodiversity across a depth gradient. *PeerJ* 7:e6379.
- Egan, S., Harder, T., Burke, C., Steinberg, P., Kjelleberg, S., & Thomas, T. (2013) The seaweed holobiont: understanding seaweed–bacteria interactions. *FEMS Microbiol. Rev.*, 37462–476.
- Keenan, K., McGinnity, P., Cross, T.F., Crozier, W.W. and Prodöhl, P.A. (2013) diveRstity: An R package for the estimation and exploration of population genetics parameters and their associated errors. *Methods Ecol. Evol.*, 4: 782–788.
- Koziol, A., Stat, M., Simpson, T., *et al.* (2019) Environmental DNA metabarcoding studies are critically affected by substrate selection. *Mol. Ecol. Resour.*; 19: 366– 376.
- Noda, M., Ohara H., Murase, N., Ikeda, I., Yamamoto, K.-I. (2014). The grazing of *Eisenia bicyclis* and several species of Sargassaceous and Cystoseiraceous seaweeds by *Siganus fuscescens* in relation to the differences of species composition of their seaweed beds. *Nippon Suisan Gakkaishi*, 80: 201–213.
- Raj, A., Stephens, M., and Pritchard, J.K. fastSTRUCTURE: variational inference of population structure in large SNP data sets. *Genetics*, 197: 573-589.
- Takahashi, M., DiBattista, J.D., Jarman, S. *et al.* (2020) Partitioning of diet between species and life history stages of sympatric and cryptic snappers (Lutjanidae) based on DNA metabarcoding. *Sci. Rep.*, **10**: 4319.
- Sundqvist, L., Keenan, K., Zackrisson, M., Prodöhl, P., & Kleinhan, D. (2016) Directional genetic differentiation and relative migration. *Ecol. Evol.*, 6: 3461-3475.
- West, K.M., Stat, M., Harvey, E.S, *et al.* (2020) eDNA metabarcoding survey reveals fine scale coral reef community variation across a remote, tropical island ecosystem. *Mol Ecol.*, 29: 1069–1086.
- Zarco-Perello, S., Carroll, G., Vanderklift, M., Holmes, T., Langlois, T.J., & Wernberg, T. (2020) Range-extending tropical herbivores increase diversity, intensity and extent of herbivory functions in temperate marine ecosystems. *Funct Ecol.*, 00: 1– 11.

Reviewers' Comments:

Reviewer #1:

Remarks to the Author:

As mentioned in my first assessment, this work is of high quality and the paper should be of interest to a range of scientists, conservation managers and stakeholders.

I am satisfied with the level and detail of the revision, and in particular with the additional analyses of population genetics. These analyses are important since they provide empirical support to inferences made about the amount and direction of gene flow in this system.

I only have a minor comment about the sentence below, from the Abstract:

"Population genomics of this rabbitfish indicated little genetic variation between tropical residents and vagrants"

Do authors mean "little genetic differentiation"?

Overall, I commend the authors on this well executed and interesting study.

Best regards,
Luciano Beheregaray

Reviewer #2:

Remarks to the Author:

The authors have carefully and thoughtfully addressed the concerns of all three viewers. I have no further comments at this stage, and recommend advancement of this manuscript through the editorial process: Accept.

Reviewer #4:

Remarks to the Author:

I have reviewed the ms by Gajdzik et al. commenting, as requested, specifically on whether concerns of the previous Reviewer 3, regarding potential biases in the dietary DNA metabarcoding methodology and additional diet analyses, have been addressed.

I attach here the rebuttal letter, where I highlighted in red my comments where I felt that concerns of Reviewer 3 were not fully addressed by the authors.

Additional minor comments:

L348-249: "was determined by an identified using"? Please correct

L357: a maximum OF five

Climate-assisted persistence of tropical fish vagrants in temperate marine ecosystems

Laura Gajdzik^{1,2*}, Thomas M. DeCarlo², Adam L. Koziol^{1,3}, Mahsa Mousavi-Derazmahalleh¹, Megan Coghlan¹, Matthew W. Power¹, Michael Bunce¹, David V. Fairclough⁴, Michael J. Travers⁴, Glenn I. Moore^{5,6}, and Joseph D. DiBattista^{1,7}.

We are very grateful to the reviewer #1 and reviewer #2 who commended the quality of our manuscript (MS) and recommended its publication.

In this second revision of the MS, we addressed reviewer #4's concerns about the technicality of our dietary DNA metabarcoding analysis, which needed more clarification and a quick re-analysis of our results. The new results were added in the main text but confirmed about previous findings about diet versatility for the rabbitfish *Siganus fuscescens*.

All the minor comments were also taken into consideration and the text was modified accordingly. All new changes are highlighted in in the MS.

Below we respond to the comments of each referee in detail. The reviewers' comments are in black and our response after each point is in green.

Reviewer #1

As mentioned in my first assessment, this work is of high quality and the paper should be of interest to a range of scientists, conservation managers and stakeholders.

I am satisfied with the level and detail of the revision, and in particular with the additional analyses of population genetics. These analyses are important since they provide empirical support to inferences made about the amount and direction of gene flow in this system.

I only have a minor comment about the sentence below, from the Abstract:

"Population genomics of this rabbitfish indicated little genetic variation between tropical residents and vagrants"

Do authors mean "little genetic differentiation"?

We have changed the word "variation" and to "differentiation" (line 34).

Overall, I commend the authors on this well executed and interesting study.

We are very grateful to Prof. Luciano Beheregaray for the positive and constructive feedback throughout this review process.

Best regards,
Luciano Beheregaray

Reviewer #2

The authors have carefully and thoughtfully addressed the concerns of all three viewers. I have no further comments at this stage, and recommend advancement of this manuscript through the editorial process: Accept.

We are very grateful to reviewer #2 for the constructive feedback.

Reviewer #4

I have reviewed the ms by Gajdzik et al. commenting, as requested, specifically on whether concerns of the previous Reviewer 3, regarding potential biases in the dietary DNA metabarcoding methodology and additional diet analyses, have been addressed.

Please see below our reponses to reviewer #4's comments, which required further clarification and a quick re-analysis that further substantiated our original conclusions.

I attach here the rebuttal letter, where I highlighted in red my comments where I felt that concerns of Reviewer 3 were not fully addressed by the authors.

We extracted reviewer #4's comments in red from the rebuttal letter and placed them in the next section. We kept the original reviewer #3's comments in black, our original responses in blue, and our new reponses in green.

Comments from the rebuttal letter:

Methods:

#####

Reviewer #3's comment: l. 187: for non-specialists, it could be useful here to specify why the authors performed a pluriannual sampling and/or whether this could affect or not the results. Authors could also specify here if the DNA extraction was performed right after each sampling sessions or all at once. Are potential time effects accounted for in the analyses?

Authors' reponse to reviewer #3: The sampling years were added in Table S7 (where the metadata for each individual is provided) as well as in lines 322-324. However, seasonality of sampling has not been recorded systematically for all individuals. The goal of this MS is to compare the feeding ecology of pre- and post-heatwave rabbitfish individuals, and our four-year sampling design allowed us to answer this (lines 85-87).

Reviewer #4's comment: The authors state that samples were either frozen or processed soon after collection and indicate that sampling of rabbitfish was done in 2013-2017. However, I am confused on why the authors talk about pre- and post heatwave as the heatwave was in 2010/2011, so prior to sampling and they in fact compare tropical residents and vagrants to temperate environments. For the metabarcoding analysis I find their sample sizes relatively small, also considering that some samples failed. While I still recognise the value of the work, I think authors should at least mention potential limitations of their small sample sizes

Authors' response to reviewer #4: The reviewer #4 is right and our samples were not taken pre-heatwave (line 90). Apologies for this oversight. Our sample size of 7-10 individuals per locality is similar to other diet studies that also used DNA metabarcoding approaches (e.g., Brandl et al. 2020, Takahasi *et al.* 2020). To clarify, we have added two sentences in the main

text (lines 172-173) and in the methodology section (lines 330-332) about some potential limitations due to low sampling size for the tropical/subtropical locations in our study.

Reviewer #3's comment: Zinger, L. (2018). Environmental DNA: For biodiversity research and monitoring. Oxford University Press.). In this case, most of the information retrieved with the authors methodology might correspond to material that is actually difficult to digest and hence, of lesser importance in terms of source of energy. Do the authors have any information that would ensure that it is not the case (e.g. a priori known feeding habits based on observations)?

Authors' response: Using shorter primers, such as trnL, usually only allow broad taxonomic classification (phyla level or family level), whereas longer regions allow for improved taxonomic classification, which is key for our MS that investigates fine-scale dietary patterns. Additionally, *S. fuscescens* is known to consume brown (e.g., Sargassum spp., Ecklonia radiata) and red macroalgae (e.g., Dictyota spp.) as well as cyanobacteria (Egan et al. 2013; Noda et al. 2014; Zarco-Perello et al. 2019). Therefore, our choice of primers needed to be able to address this diet diversity.

Reviewer #4's comment: The point made by the reviewer is an important one and lays at the basis of DNA metabarcoding applications to degraded DNA mixtures. The choice of short metabarcodes prioritizes accurate identification from highly fragmented DNA rather than taxonomic resolution; in contrast longer barcodes would for sure be more informative taxonomically, but less efficient in amplifying short DNA fragments from degraded digested food items. So I share the concern of the reviewer and I agree that providing diet information from independent sources would allow giving support that the items identified do not merely correspond to things hard to digest.

Authors' response to reviewer #4: As per our response to reviewer #3, *Siganus fuscescens* is known to feed on red, brown, and green marine macrolagae as well as on cyanobacteria (e.g., Egan et al. 2013, Noda et al. 2014, Hyndes et al. 2017, Zarco-Perello et al. 2017). These food sources were targeted in the present study by amplifying a portion of the 23S rRNA gene isolated from rabbitfish stomach contents. We have added the references of these independent sources in the methodology part at lines 334-335.

Reviewer #3's comment: 1.296: I guess this should read instead that “sequencing reads were dereplicated”, or “we kept only one version of replicated reads as well as their associated number of replicates”. The term “merge” is inappropriate.

Authors' response to reviewer #3: This sentence has been rephrased. Please see lines 378-384.

Reviewer #4's comment: The expression now used in the revised version “dereplicate each sample down to unique sequences” is inaccurate and confusing, the authors should follow the reviewer suggestion.

Authors' response to reviewer #4: The sentence has been modified according to reviewer 3#’ suggestion: “Sequencing reads were dereplicated into unique sequences...”. Please see lines 388-391.

Reviewer #3's comment: 1.304 suggests that the authors removed any sequence occurring in the negative controls. While at first glance, this strategy sounds reasonable, the data produced by the authors may be polluted with “tag-jumps” (see e.g. Esling, P., Lejzerowicz, F., & Pawlowski, J. (2015). Accurate multiplexing and filtering for high-throughput amplicon-sequencing. Nucleic Acids Research, 43(5), 2513-2524). This can result in the occurrence of very abundant and biologically meaningful OTUs in negative controls. Did the authors ensure that this bias is limited/absent in their data (e.g. by looking if they obtained sequences with unexpected MID combinations)? Was LULU used to reduce this bias? If none of the above was considered,

then removing OTUs in the negative controls could actually remove important OTUs from the dataset.

Authors' response to reviewer #3: The sequences of the OTUs appearing in the control samples have been blasted and assigned to sequences that belonged to rice and terrestrial plants. We have provided additional details on these assignments at lines 393-397. Additionally, this is common practice in DNA metabarcoding paper to remove sequences from the negative control (e.g., West et al. 2020).

Reviewer #4's comment: The authors responded only partially to the reviewer. It's good that they blasted the OTUs found in the negative controls, but they did not provide any information on if and how they accounted for the problem of tag-jumping in their data which is a real concern in DNA metabarcoding studies. In addition to extraction negative controls, other types of controls including PCR negative and positive controls and empty PCR wells are now common practice in DNA metabarcoding studies to monitor the experiment and set filtering parameters. Also it seems that PCRs were performed in duplicate, but that their data were simply combined and not used for identifying erroneous/spurious/contaminant sequences. Such measures have become standards and should be applied in DNA metabarcoding studies for quality assurance and allow robust inferences. See Zinger, L., Bonin, A., Alsos, I. G., Bálint, M., Bik, H., Boyer, F., ... Taberlet, P. (2019). DNA metabarcoding—Need for robust experimental designs to draw sound ecological conclusions. *Molecular Ecology*, 28(8), 1857–1862. doi: [10.1111/mec.15060](https://doi.org/10.1111/mec.15060).

Authors' response to reviewer #4: As mentioned by reviewer #3, if tag-jumping was indeed a problem, a high number of biologically meaningful OTUs would have appeared in the negative controls, which were uniquely tagged as advised by Zingler et al. (2019). However, this was not the case since the 17 OTUs present in our negative controls were not assigned to any marine macroalgae and zooplankton, the targets of our study. In addition to blanks, we also follow recommendations from Schnell, Bowman, and Gilbert (2015): (i) we designed tags with a high number of minimum differences (8 nucleotide-long tags with a minimum of 2 nucleotide difference for all samples), (ii) we worked in completely separate pre- and post-PCR laboratories (PCR master mixes were even prepared in an ultra-clean lab where no DNA samples are handled and by using the automated QIAcube robot we reduce human error), (iii) we handled primers, amplicons, and libraries as appropriate, (iv) we only merged 100% identical paired reads, and (v) we did not perform post-ligation PCR enrichment, and (vi) we did not use multiple forward and reverse tag combinations during qPCR to reduce the chimera formation. Nevertheless, we reran the analysis and increased the abundance threshold to 10 read sequences per OTU per individual sample. The results of this more conservative analysis did not change the main findings of diet versatility but rather substantiate our original conclusions. Please see lines 158-169, lines 412-415, and Figs 4- 5.

Although we did not use a positive control during the amplification of the rabbitfish stomach contents due to a lack of access to stomachs from other herbivorous fishes, we did successfully amplify the 23S rRNA gene from voucher tissues of the kelp forest *Ecklonia radiata*, indicating that the 23S rRNA primer used in this study can successfully amplify this and related species. We consider this to be a proxy for a positive control.

Concerning the use of empty wells, we are not sure what the reviewer #4 is referencing to and thus cannot address the reviewer's concern.

Finally, pooling each PCR sample that were conducted in duplicate at equimolar ratio to form a library is common practice (e.g., Koziol et al. 2018, Boggs et al. 2019, West et al. 2020a,b). Unlike what the reviewer #4 suggested, we have conducted thorough analyses to identify erroneous and spurious sequences. We have removed short reads, chimeras, and erroneous OTUs caused by co-occurrence error through the use of a post-clustering filtering procedure with the R-package Lulu (lines 384-400).

Reviewer #3's comment: 1.318: specify how these were standardized.

Authors' response to reviewer #3: This sentence has been rephrased (lines 411-414).

Reviewer #4's comment: L412: Total number of sequences or of sequence reads?

Authors' response to reviewer #4: We calculated the relative abundance of each food item per individual stomach, which corresponds to the number of sequence reads of each food item divided by the total number of sequence reads of each individual stomach sample. We have added the words "...reads.." and "...individual stomach content.." for further clarity in the main text (lines 421-424).

Reviewer #3's comment: 1.327: The Indval species test is useful to discover indicator species of a given condition, but it is not a proper way to assess whether the diet composition is the same or not in the different locations, it's rather a complementary analysis. The authors should first consider performing a PERMANOVA or a related test (on abundance or presence absence data). In addition, the Indval species metric is also sensitive to rare OTUs (e.g. those occurring in only one sample), are such OTUs included in this analysis?

Authors' response to reviewer #3: As mentioned before, we have performed three additional statistical analyses to address these issues: nMDS, PERMANOVA, and PERMDISP2.

Reviewer #4's comment: OK, but authors do not answer the last question from the reviewer about the inclusion of rare OTUs

Authors' response to reviewer #4: In the final OTU table, 15 taxonomically-assigned OTUs out of 78 OTUs appear a single time in one sample, but each of these OTUs is based on 10-322 sequences. Therefore, these OTUs most likely represent biologically-meaningful food items that need to be accounted for in the IndVal analysis. However, their influence was reduced by using the relative abundance approach, instead of a presence/absence dataset, which decreases the weight of these rare OTUs compared to the most abundant OTUs. This is one of the most conservative methods of exploring relative abundance of dietary items.

Additional minor comments:

L348-249: "was determined by an identified using"? Please correct
"an identified" was a mistake and should have been removed from the sentence. The text has been modified accordingly (line 357).

L357: a maximum OF five
The word "of" has been added (line 365).

References

Boggs, L.M., Scheible, M.K.R., Machado, G., Meiklejohn, K.A. (2019) Single fragment or bulk soil DNA metabarcoding: which is better for characterizing biological taxa found in surface soils for sample separation? *Genes*, **10**: 431.

Brandl, S.J., Casey, J.M., & Meyer, C.P. (2020) Dietary and habitat niche partitioning in congeneric cryptobenthic reef fish species. *Coral Reefs*, **39**: 305–317.

Egan, S. *et al.* (2013) The seaweed holobiont: understanding seaweed–bacteria interactions. *FEMS Microbiol. Rev.* **37**: 462–476.

Hyndes, G. A. *et al.* (2016) Accelerating tropicalization and the transformation of temperate seagrass meadows. *Bioscience* **66**: 938–948.

Koziol, A, Stat, M, Simpson, T, et al. Environmental DNA metabarcoding studies are critically affected by substrate selection. *Mol. Ecol. Resour.*, **19**: 366–376.

Noda, M., Ohara, H., Murase, N., Ikeda, I. & Yamamoto, K.-C. (2014) The grazing of *Eisenia bicyclis* and several species of *Sargassaceous* and *Cystoseiraceous* seaweeds by *Siganus fuscescens* in relation to the differences of species composition of their seaweed beds. *Nippon Suisan Gakkaishi* **80**: 201–213.

Schnell I.B., Bohmann K., Gilbert M.T.P. (2015) Tag jumps illuminated—reducing sequence-to-sample misidentifications in metabarcoding studies. *Mol. Ecol. Resour.*, **15**: 1289–1303.

Takahashi, M., DiBattista, J.D., Jarman, S. *et al.* (2020) Partitioning of diet between species and life history stages of sympatric and cryptic snappers (Lutjanidae) based on DNA metabarcoding. *Sci. Rep.*, **10**: 4319.

West, K.M., Stat, M., Harvey, E.S., *et al.* (2020) eDNA metabarcoding survey reveals fine-scale coral reef community variation across a remote, tropical island ecosystem. *Mol Ecol.* **29**,1069–1086.

West, K.M., Richards, Z.T., Harvey, E.S. *et al.* (2020) Under the karst: detecting hidden subterranean assemblages using eDNA metabarcoding in the caves of Christmas Island, Australia. *Sci. Rep.*, **10**: 21479.

Zarco-Perello, S., Wernberg, T., Langlois, T. J. & Vanderklift, M. A. (2017) Tropicalization strengthens consumer pressure on habitat-forming seaweeds. *Sci. Rep.* **7**: 820.

Reviewers' Comments:

Reviewer #4:

Remarks to the Author:

I have reviewed this revised version of the ms. The authors have addressed almost all my concerns. I appreciate the extra work done for documenting the reliability of their results. I recommend the ms for publication.

I just wanted to clarify to the authors what I meant when I referred to Zinger et al. 2019 on the use of positive controls, controls for tag-jumping, and replication in one of my comments.

Zinger et al. 2019 talk about "tagging system negative controls", which are made by leaving some tag combinations unused (this is what I called empty PCR wells before, I apologize for the confusion). During data analysis it can be checked if these combinations are found in the resulting sequencing data. The PCR negative controls (PCR reagents without DNA) used in this study are not equivalent to these "tagging system negative controls".

Checking that the primers used in this study can successfully amplify the species of interest does not replace the use of positive controls. Positive controls are meant to be used during all steps of the metabarcoding process together with the collected samples, because this allows to evaluate the effectiveness of the metabarcoding experiment, distinguish contamination from true ecological signal, and the effectiveness of the bioinformatics filtering. A positive control can be easily artificially created in the lab, for example by mixing known DNA quantities of different taxa.

I do not question the practice of pooling replicates within the same library, rather the fact that information from independent replication was not used to verify repeatability in sequence detection. Independent technical replicates (i.e. independent PCR replicates) serve this purpose.